# Mind Dreamer: Untethering Imagination via Active Causal Intervention on Latent Manifolds

**Shaojun Xu** [1]  **Xiaoling Zhou** [2]  **Yihan Lin** [3]  **Yapeng Meng** [1 4]  **Xinglong Ji** [1]  **Luping Shi** [1]  **Rong Zhao** [1]

## Abstract

Model-Based Reinforcement Learning yields sample efficiency via latent imagination, yet remains constrained by **Historical Tethering**: imagination is typically initialized from observed states. This creates a learning asymmetry, where the world model's manifold discovery outpaces the policy's sparse-reward optimization. We propose **Mind Dreamer (MD)**, a framework that instantiates **Active Causal Intervention** to transcend Markovian continuity. MD reformulates discovery as the minimization of a global Relay Expected Free Energy. Instead of initializing from historical data, it draws initial states from an adversarial generator $s_0 \sim p_{gen}(\cdot)$, creating non-continuous **latent jumps** to epistemic blind spots that are physically plausible yet cognitively challenging. We derive **Relay Value Function** and **Relay Uncertainty Function** to resolve the credit assignment paradox across these spatial ruptures. Treating synthesized anchors as interventional intermediary states, these potentials propagate pragmatic and epistemic value through Bellman-style backups. Notably, we prove that uncertainty propagation across discontinuities necessitates a quadratic discount $\gamma^2$, establishing a formal epistemic horizon. Theoretically, MD approximates a variance-minimizing importance sampler that expands the manifold's spectral gap, reducing the hitting time to critical bottleneck states. Empirically, MD achieves a **1.67×** **average speedup** over DreamerV3 on DeepMind Control Suite, reaching **8.8×** in sparse-reward tasks.

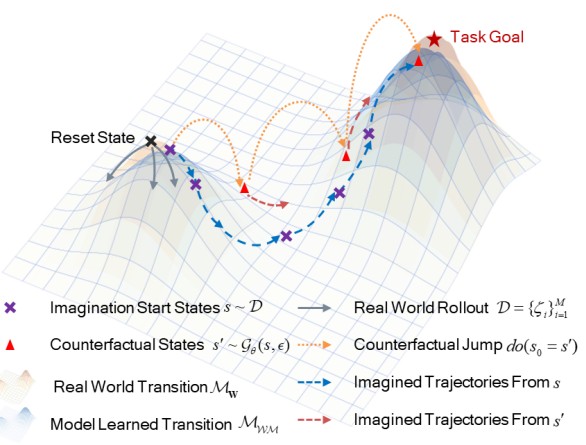

**Figure 1. Untethering Imagination via Active Causal Intervention**. Unlike standard MBRL (blue) which is **tethered** to historical observations ($s \sim \mathcal{D}$), Mind Dreamer enables proactive latent intervention beyond historical support. An adversarial generator $\mathcal{G}$ performs non-continuous **latent jumps** via generated interventional states (orange) to synthesize anchors (red). These anchors bridge OOD regions, enabling goal-directed imagination on the learned manifold.

---

[1]Center for Brain-Inspired Computing Research, Department of Precision Instrument, Tsinghua University, Beijing, China [2]College of Computer Science and Technology, Zhejiang University, Hangzhou, China [3]Pen-Tung Sah Institute of Micro-Nano Science and Technology, Xiamen University, Xiamen, China [4]Primevision Technology, Shanghai, China. Correspondence to: Rong Zhao <r_zhao@tsinghua.edu.cn>.

*Proceedings of the 43rd International Conference on Machine Learning*, Seoul, South Korea. PMLR 306, 2026. Copyright 2026 by the author(s).

## 1. Introduction

Model-Based Reinforcement Learning (MBRL) has emerged as a cornerstone of sample-efficient AI, predicated on the power of *latent imagination*—the ability to simulate future outcomes within a learned world model (Ha & Schmidhuber, 2018; Sutton, 1991; Hafner et al., 2025). This paradigm is conceptually anchored by the **Manifold Hypothesis**: while sensory observations are high-dimensional, they reside on a low-dimensional manifold $\mathcal{M}$ governed by the environment's underlying physics (Bengio et al., 2013). Under this view, MBRL is a dual-process of geometric discovery: learning the tangent space of $\mathcal{M}$ (dynamics) and regressing a value functional over its topology.

However, a fundamental bottleneck remains: **Historical Tethering**. While modern world models rapidly recover the global structure of $\mathcal{M}$ via dense self-supervised objectives (LeCun et al., 2022; Bardes et al., 2021), imagination remains a "prisoner of history"—typically initialized

only from observed states in a replay buffer (Ecoffet et al., 2019; Dabney et al., 2021). This creates a critical **Learning Asymmetry**: the agent's internal atlas of the environment's physics (the manifold) expands far more rapidly than its policy can optimize for sparse rewards (Baker et al., 2022; Sekar et al., 2020). Consequently, imagination is confined to previously traversed regions, forcing the agent into a costly random walk when encountering out-of-distribution (OOD) regions, even if its world model already possesses the structural knowledge to bridge them.

Prior attempts to mitigate exploration bottlenecks have largely relied on intrinsic motivation (Sekar et al., 2020; Houthooft et al., 2016) or goal-relabeling (Andrychowicz et al., 2017). While these methods incentivize the discovery of novel regions, they remain trajectory-bound. In this work, we operationalize this asymmetry via **Active Causal Intervention (ACI)**. We re-envision the world model as an adversarial generator capable of targeted latent intervention rather than a passive simulator By sampling initial imagination states from a learned generator $s_0 \sim p_{gen}(\cdot)$ rather than the historical buffer $s_0 \sim \mathcal{D}$, ACI transcends the constraints of Markovian continuity, enabling the agent to perform non-continuous **latent jumps** to synthesized anchors at the frontiers of its knowledge.

We introduce Mind Dreamer (MD), a framework that transforms MBRL from a passive replay mechanism into an information-theoretic stress test. MD utilizes an adversarial state generator $\mathcal{G}$ that lifts the principle of Expected Free Energy (EFE) (Friston, 2009; Tschantz et al., 2020) from a local policy-selection metric to a global discovery functional. This allows the agent to identify intermediary states—regions where pragmatic goal-alignment and epistemic uncertainty are optimally balanced (Friston et al., 2017b). However, evaluating a non-local jump presents a *credit assignment paradox*: temporal consistency is broken. To resolve this, we derive two recursive functionals: the **Relay Value Function (RVF)** and **Relay Uncertainty Function (RUF)**. Unlike standard value functions that treat states as destinations, these Relay Potentials treat synthesized anchors as **intermediary states**, providing a Bellman-style formulation to evaluate the multi-step utility of a jump across spatial ruptures.

Our theoretical analysis demonstrates that Mind Dreamer approximates a **variance-minimizing importance sampler** for manifold refinement. We prove that minimizing the R-EFE objective is mathematically equivalent to minimizing the variance of the world model's gradients, effectively performing a "Manifold Repair" curriculum. Crucially, we show that integrating informational shocks requires a **quadratic discount** $\gamma^2$, establishing a formal *Epistemic Horizon* that prevents the generator from chasing distal hallucinations. From a topological perspective, we prove that

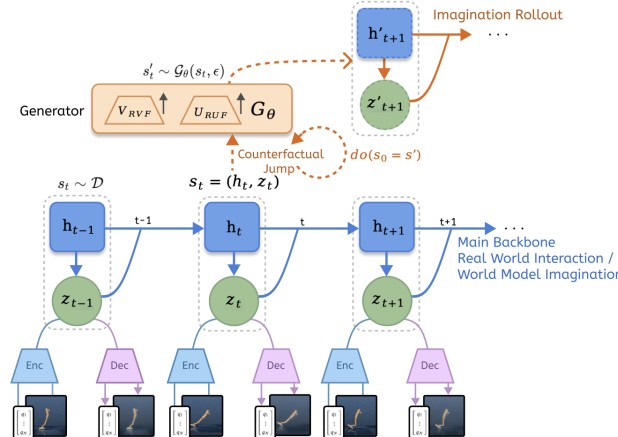

*Figure 2.* **The Mind Dreamer Paradigm.** Unlike standard MBRL tethered to historical observations, MD enables proactive discovery via: (1) **Adversarial Synthesis**, where a generator $\mathcal{G}_\theta$ uses generated initial states to create interventional anchors $s'$ at manifold frontiers. (2) **Relay Guidance**, which leverages the Relay Value ($V_{RVF}$) and Uncertainty ($V_{RUF}$) functions to assign potential across spatial ruptures. This untethers imagination from past trajectories, targeting physically plausible yet epistemically rich regions for global manifold repair.

MD increases the manifold's *conductance*, effectively expanding the manifold's spectral gap, reducing the hitting time to critical states.

Our contributions are summarized as follows:

**A New Paradigm: Active Causal Intervention (ACI).** We identify the "Historical Tethering" limitation and introduce the latent generator distribution $p_{gen}$ to enable goal-directed, non-continuous intervention.

**Bridging Discontinuity: Relay Potential Fields.** We derive RVF and RUF as recursive functionals as credit assignment to bridge spatial ruptures, introducing the $\gamma^2$ discount for stable uncertainty propagation across **latent jumps**.

**Theoretical Optimality and Speedup.**

We prove that MD performs optimal importance sampling on manifold residuals, yielding a convergence speedup proportional to the expansion of the manifold's spectral gap.

**Empirical Validation.**

On the DeepMind Control Suite, Mind Dreamer achieves a **1.67× average speedup** in sample efficiency over DreamerV3, with acceleration reaching **up to 8.8×** in challenging bottleneck tasks.

## 2. Related Works

The evolution of MBRL has transitioned from simple Dyna-style planning (Sutton, 1991) to complex latent imagination (Ha & Schmidhuber, 2018; Hafner et al., 2023). However, the reliance on historical trajectory support remains a bottleneck. Mind Dreamer sits at the intersection of Active Inference, Causal Intervention, and Manifold Topology.

### 2.1. From Passive Curiosity to Active Manifold Repair

Traditional exploration, such as prediction error (Pathak et al., 2017) or information gain (Houthooft et al., 2016) often relies on intrinsic bonuses to steer the agent's random walk. While recent active inference works (Tschantz et al., 2020; Mazzaglia et al., 2021; Millidge, 2020; Friston et al., 2017b; Çatal et al., 2020) treat control as inference, unified exploration and exploitation under Free Energy Principle (Friston, 2009), they remain *trajectory-bound*: Expected Free Energy (EFE) is typically evaluated along continuous Markovian paths. Plan2Explore (Sekar et al., 2020) optimizes for global curiosity but is still constrained by initializing rollouts from the replay buffer. Mind Dreamer (MD) transcends this by lifting EFE into a global curriculum for **Manifold Repair**, using an adversarial generator to synthesize interventional anchors in regions where the world model's geometry is most "brittle."

### 2.2. Beyond Trajectory Exploitation: Latent $do$-Intervention

Standard techniques like Hindsight Experience Replay (HER) (Andrychowicz et al., 2017) and Hierarchical RL (Nachum et al., 2018; Li et al., 2020) excel at trajectory exploitation—maximizing the utility of *collected* data through relabeling or abstraction. However, they remain "prisoners of the buffer," unable to reason beyond the convex hull of past experiences. Our approach shifts the paradigm toward **Manifold Intervention** (Pearl, 2009). By sampling from a learned intervention distribution $p_{gen}$, MD decouples environmental physics from historical policy bias. Unlike Go-Explore (Ecoffet et al., 2021), which requires explicit environment resets to known states, MD performs mental teleportation to synthesized frontiers, performing intervention-based exploration in unvisited latent regions.

### 2.3. Credit Assignment Across Spatial Ruptures

Breaking Markovian continuity presents a significant credit assignment paradox. Standard temporal difference methods like TD($\lambda$) (Sutton, 1988) or Eligibility Traces (Sutton & Barto, 2018; Seijen & Sutton, 2014) fail when trajectories are non-continuous. While Successor Representations (SR) (Dayan, 1993; Machado et al., 2018) decouple dynamics from rewards, they typically require continuous

discovery to build the occupancy map. MD extends this logic via **Relay Potential Fields**. Unlike Goal-Conditioned RL (Schaul et al., 2015) where targets are terminal sinks, our RVF and RUF reformulate synthesized states as **interventional intermediary states**. This allows the agent to propagate pragmatic and epistemic value across spatial ruptures, identifying high-leverage via-points that connect disjoint regions of the manifold.

## 3. Methodology: Mind Dreamer

The core philosophy of MD is that efficient learning requires the agent to transcend its historical trajectories and proactively explore unmastered regions of the latent manifold $\mathcal{M}$ through structured **latent jumps**.

### 3.1. Problem Formulation: MBRL on Latent Manifolds

We reformulate the standard POMDP through the lens of the Manifold Hypothesis (Cayton et al., 2005), by assuming that observations $o \in \mathcal{X}$ are realizations of an underlying low-dimensional latent manifold $\mathcal{M} \subset \mathbb{R}^d$, where $d \ll \dim(\mathcal{X})$, and an encoder $e_\psi : \mathcal{X} \to \mathcal{M}$ maps observations to latent states $s \in \mathcal{M}$. This perspective allows us to decompose MBRL into two distinct learning processes:

**Learning the Transition Structure (World Model)**: The world model $p_\psi(s_{t+1}|s_t, a_t)$ approximates the transition dynamics. By minimizing reconstruction and transition errors, the model learns the local transition structure of the manifold.

**Learning the Value Field (Value Function)**: The optimal value function $V_\phi^*(s)$ represents a scalar field over $\mathcal{M}$ that quantifies task-optimal potential.

### 3.2. Active Causal Intervention

To transcend the "Historical Tethering" of standard MBRL, we propose ACI. This mechanism shifts the world model's role from a passive simulator of past experiences to an active adversarial generator for directed manifold discovery. The core of ACI is leveraging the Learning Asymmetry: the world model captures the transition structure of the latent manifold $\mathcal{M}$ via dense self-supervised signals far more rapidly than the policy resolves sparse rewards. We exploit this by introducing a learned intervention distribution $p_{gen}$.

**Definition 3.1. Causal Intervention Jump**. Sampling $s_0 \sim p_{gen}(\cdot)$ allows the agent to initialize imagination at any anchor $s' \in \mathcal{M}$, provided $s'$ satisfies the manifold consistency $\mathcal{L}_{mf}$ (see Sec. 3.5). This decouples discovery from historical buffer $\mathcal{D}$ while remaining anchored to the Markovian flow $s_t \sim p_\psi(\cdot|s_{t-1}, a_{t-1})$ from learned physics of the transition structure.

Unlike standard rollouts initialized from the replay buffer $s_0 \sim \mathcal{D}$, ACI allows the agent to sample $s_0 \sim p_{gen}(\cdot)$, where $s' \in \mathcal{M}$ is a intervention anchor synthesized by an adversarial generator $\mathcal{G}_\theta(s, \epsilon)$. This allows the imagination to "teleport" across spatial ruptures, investigating regions that are physically consistent with the environment's laws but have not been mastered by the current policy. The generator $\mathcal{G}$ acts as an adversarial stress-tester that identifies regions where the latent manifold's transition structure $T\mathcal{M}$ is under-sampled or where the value function $V_\phi$ exhibits high curvature. $\mathcal{G}_\theta(s, \epsilon)$ operates exclusively during the imagination phase, ensuring that no extra interaction costs are imposed on the environment.

### 3.3. From Local EFE to Global Relay Discovery

To guide the state generator $\mathcal{G}$, we leverage the framework of Active Inference, replacing simple reward maximization with the minimization of Expected Free Energy (EFE) (Friston et al., 2017b). While standard MBRL is reactive, EFE provides a proactive objective that unifies goal-directed behavior (pragmatic value) with curiosity-driven exploration (epistemic value). In App. A.2, we demonstrate that MaxEnt RL is a special case of EFE (Millidge et al., 2020).

**Definition 3.2. The Local Objective.** For a synthesized anchor $s'$, the local EFE $G(s')$ quantifies its immediate utility (inference details in Appendix A.1):

$$G(\pi, \tau) = -\beta \underbrace{\mathcal{I}(s_\tau; o_\tau | \pi)}_{\text{Epistemic Value } \mathcal{E}(s')} -\eta \underbrace{\mathbb{E}_{q(o_\tau)}[\ln p(o_\tau)]}_{\text{Pragmatic Value} \mathcal{P}(s')} \quad (1)$$

where $\mathcal{P}(s')$ aligns with the agent's goal-prior $p(o|C)$ (Levine, 2018) and $\mathcal{E}(s')$ measures the Mutual Information $\mathcal{I}$, identifying regions where the world model's tangent space is ill-defined (Houthooft et al., 2016).

However, Point-wise EFE fails for non-local jumps because it ignores the **reachability** and **distal potential** beyond the jump. We therefore lift $G(s')$ into the Relay EFE (R-EFE), denoted as $\Psi(s, s')$.

$$\Psi(s, s') = \mathbb{E}_q \left[ \sum_{k=1}^{H} \gamma^k G(s_k) \middle| s_0 = s, s' \in \xi \right] \quad (2)$$

R-EFE represents a path-integral of free energy, evaluating not just the immediate utility of an anchor $s'$, but the cumulative potential of all trajectories branching from $s'$.

### 3.4. Bridging the Gap: Relay Potential Fields

Evaluating the R-EFE objective across spatial ruptures presents a significant credit assignment challenge: as standard value functions depend on local temporal consistency $(s_t \rightarrow s_{t+1})$, they fail to propagate gradients across non-continuous spatial ruptures $(s_t \rightarrow s'_{t+1})$. We decompose

the Eq. 2 into two recursive **Relay Potential Fields**. These fields act as tractable proxies, allowing the generator $\mathcal{G}$ to perform functional gradient ascent on the manifold's utility landscape.

**Pragmatic Proxy (RVF)**: Unlike goal-conditioned RL typically treats $s'$ as a terminal destination (Andrychowicz et al., 2017; Schaul et al., 2015), Relay Value Function $V_{\text{RVF}}$ treats the anchor $s'$ as a **interventional intermediary state** rather than a terminal destination. It integrates the cost of reaching $s'$ with the value function $V_\phi(s')$ available thereafter (Formal definition in Appendix C.2):

$$V_{RVF}(s, s') \triangleq \max_\pi \mathbb{E}_{\substack{\pi \\ s_k = s'}} \left[ \sum_{t=0}^{k-1} \gamma^t r_t + \gamma^k V_\phi(s') \right] \quad (3)$$

This ensures $\mathcal{G}$ identifies Pragmatic Gateways: states that are not necessarily high-reward themselves but are strategically positioned to unlock high-density reward regions.

**Epistemic Proxy (RUF)**: The Relay Uncertainty Function $V_{\text{RUF}}$ quantifies the **informational density** of a jump. Crucially, it aggregates expected information gain $\mathcal{I}$ using a **quadratic discount** $\gamma^2$ (Derivation in Appendix B.3):

$$V_{RUF}(s, s') \triangleq \mathbb{E}_{\substack{\pi \\ s_k = s'}} \left[ \sum_{t=0}^{k-1} \gamma^{2t} \mathcal{I}_{t+1} + \gamma^{2k} U_{\phi_u}(s') \right] \quad (4)$$

Where $U_{\phi_u}(s')$ is the Bellman Uncertainty Function (O'Donoghue et al., 2018a). As detailed in 4.4, the $\gamma^2$ term is not a hyperparameter but a requisite for stable variance propagation, ensuring the generator targets regions of genuine epistemic volatility while remaining robust to distal hallucinations where model variance compounds uncontrollably.

### 3.5. The Adversarial Curriculum: Structural Constraints and Co-evolution

By redefining EFE minimization through RVF and RUF maximization, we transform the active inference problem into a standard RL optimization task for the generator $\mathcal{G}_\theta(s, \epsilon)$:

$$\max_\theta \mathbb{E}_{\substack{s \sim \mathcal{D} \\ s' \sim \mathcal{G}_\theta(s, \epsilon)}} \left[ \eta V_{RVF}(s, s') + \beta V_{RUF}(s, s') - \lambda \mathcal{L}_{mf}(s') \right] \quad (5)$$

This creates a adversarial minimax optimization: $\mathcal{G}$ proactively identifies the weakest links (high RUF) or shortest paths (high RVF) in the agent's current atlas. To prevent $\mathcal{G}$ from generating latent hallucinations, we constrain it to the manifold learned by the world model $p_\psi(s_{t+1}|s_t, a_t)$ through structural self-consistency. The manifold constraint $\mathcal{L}_{mf}$ includes:

**Dynamics Coherence**: Transition entropy $\mathcal{H}(p_\psi(\cdot|s', a))$ is penalized to prevent $\mathcal{G}$ from exploiting ill-defined "cracks"

in the world model's dynamics. This effectively defines a "Pessimistic Trust Region" where the model's predictions remain valid (Kidambi et al., 2020).

**Cycle-Consistency**: Enforces $D_{KL}[\text{Enc}(\text{Dec}(s'))\|s']$ to anchor jumps within the verifiable reconstruction manifold (Zhu et al., 2017), ensuring that synthesized states remain representable by the World Model. This constraint ensures that $\mathcal{L}_{mf}$ acts as a trust-region boundary for the adversarial generator (Schulman et al., 2015).

**Generator-Policy Co-evolution**. As $\mathcal{G}$ shifts the imagination distribution, the policy $\pi$ must remain stable. We employ quasi-static Target Potential Networks for $V_{RVF}$ and $V_{RUF}$ to ensure non-recursive bootstrap stability (Mnih et al., 2015), preventing the divergence typically seen in off-policy estimation on non-linear manifolds. Furthermore, the World Model $p_\psi$ is updated at a lower frequency relative to the generator $\mathcal{G}$, providing a more stationary transition landscape for state synthesis (Heusel et al., 2017).

While optimizing Eq. 5 implies direct gradient ascent on the potential functions, doing so with neural approximators can lead to value overestimation. In practice, we stabilize the adversarial training by casting $\Psi$ as an energy function within an InfoNCE contrastive objective, forcing the generator to produce states that exhibit strictly higher potential than a dynamic baseline of historical experiences.

# 4. Theoretical Analysis: From Local Actions to Global Manifold Interventions

We formally ground Mind Dreamer (MD) by demonstrating that RVF and RUF are not merely heuristics but represent the decomposition of Expected Free Energy (EFE) into recursive functionals, and that "jumping" to high-EFE states is a kind of optimal sampling on the global manifold that accelerates manifold refinement and optimal value function convergence.

## 4.1. Operator Foundations: Mapping EFE to Relay Potentials

We first establish that RVF and RUF are not merely heuristics but represent the decomposition of Expected Free Energy (EFE) into recursive functionals.

**Definition 4.1. Relay Potential Operators**. Let $s$ be an anchor and $s'$ an intervention jump. Given $\tau_{s'} = \inf\{t \geq 0 : s_t = s'\}$ as the first hitting time of $s'$, we define the Pragmatic Relay Operator $\mathcal{T}_V$ and Epistemic Relay Operator $\mathcal{T}_U$ as:

$$(\mathcal{T}_V V)(s, s') = \mathbb{E}_\pi \left[ \sum_{t=0}^{\tau_{s'}-1} \gamma^t r_t + \gamma^{\tau_{s'}} V_\phi(s') \right] \quad (6)$$

---

**Algorithm 1** Mind Dreamer: Active Causal Intervention

1: **Initialize:** World Model $\mathcal{WM}$ (Encoder $e_\psi$, RSSM $p_\psi$), Value $V_\phi$, Uncertainty $U_{\phi_u}$, Policy $\pi_\omega$, Generator $\mathcal{G}_\theta$, Relay Function $V_{RVF}$ and $V_{RUF}$, Buffer $\mathcal{D}$
2: **for** each training step **do**
3:  1. **World Model Learning**: Update $\{e, p\}$ using $\mathcal{D}$ (Standard RSSM loss)
4:  **2. Mind Dreamer**
5:  Sample $s_0 \sim \mathcal{D}$
6:  $s' \leftarrow \mathcal{G}_\theta(s_0, \epsilon)$ where $\epsilon \sim \mathcal{N}(0, I)$
7:  Sample negative pool $s_{neg} \sim \mathcal{D}_{buffer} \cup \mathcal{D}_{elite}$
8:  Compute EFE Scores: $\Psi(s') \leftarrow \eta V_{RVF}(s') + \beta V_{RUF}(s')$ and $\Psi(s_{neg})$
9:  $\mathcal{L}_{contrast} \leftarrow \max(0, m - (\Psi(s') - \max \Psi(s_{neg})))$ {InfoNCE Surrogate}
10:  $\mathcal{L}_{mf} \leftarrow \mathcal{H}(p_\psi(\cdot|s', a)) + D_{KL}[\text{Enc}(\text{Dec}(s'))\|s']$
11:  Update $\theta \leftarrow \nabla_\theta (\mathcal{L}_{contrast} + \lambda \mathcal{L}_{mf})$ {Stable Adversarial Optimization}
12:  3. **Policy Optimization**:
13:  Start imagination rollouts from generated $z_{s'}$ (posterior features of $s'$)
14:  Update $\pi$, $V_\phi$ and $U_{\phi_u}$ via TD-learning
15:  Update $V_{RVF}$ and $V_{RUF}$ via $k$-horizon HER with non-recursive bootstrap target (Eq. 3 and Eq. 4)
16:  **4. Environment Interaction**
17:  Execute $\pi_\omega$ in real environment, collect data to $\mathcal{D}$
18: **end for**

---

$$(\mathcal{T}_U U)(s, s') = \mathbb{E}_\pi \left[ \sum_{t=0}^{\tau_{s'}-1} \gamma^{2t} \mathcal{I}_{t+1} + \gamma^{2\tau_{s'}} U_{\phi_u}(s') \right] \quad (7)$$

**Theorem 4.2. *Path-Integral Equivalence*.** *The fixed points $V_{RVF}^*$ and $V_{RUF}^*$ uniquely recover the $k$-step path-integrated constituents of the Expected Free Energy (EFE)* (Kappen, 2005).

*Proof Sketch*: Under an instrumental prior $p(o|C) \propto \exp(V^*(s)/\lambda)$, maximizing $V_{RVF}$ minimizes cumulative pragmatic risk (see Appendix C.1), while under Gaussian posterior approximations (Millidge et al., 2021) $V_{RUF}$ providing a recursive measure of epistemic leverage across manifold ruptures, maximizing $V_{RUF}$ is equivalent to maximizing the expected reduction in model surprisal (Friston et al., 2015; Houthooft et al., 2016), where informational shocks $\mathcal{I}$ propagate with a quadratic discount $\gamma^2$.

**Lemma 4.3. *Contraction and Uniqueness*.** *On a connected latent manifold $\mathcal{M}$, for any $s' \neq s$, the operator $\mathcal{T}_V$ and $\mathcal{T}_U$ is a $\gamma$ and $\gamma^2$ contraction mapping under the $L_\infty$ norm.*

*Proof Sketch*: Since $\tau_{s'} \geq 1$ for any non-trivial jump, the discount factor applied to the bootstrap term $V_\phi(s')$ satisfies $\gamma^{\tau_{s'}} \leq \gamma < 1$. This ensures that $\mathcal{T}_V$ obeys the Banach

Fixed-Point Theorem, admitting a unique fixed point $V_{RVF}^*$ that represents the maximum potential of all policy flows passing through the gateway $s'$. The same applies to $\mathcal{T}_U$, guaranteeing a stable global landscape for the state generator. Detailed proof in Appendix C.4

*Remark* 4.4. **The Epistemic Horizon**: The $\gamma^2$ discount in RUF (Eq. 4) is not an empirical tuning parameter but a necessity of variance propagation. While pragmatic rewards propagate via $\gamma$, epistemic informational shocks decay quadratically (O'Donoghue et al., 2018b) (Derivation in Appendix B.3). This quadratic decay ensures that the generator ignores "distal hallucinations" where compounding model variance renders information unverifiable.

*Remark* 4.5. **Robustness to Aleatoric Noise.** A higher-order discount ($\gamma^3$) is not required to handle aleatoric noise. The $\gamma^2$ discount strictly follows the variance operator property ($\text{Var}(\sum \gamma^t \epsilon_t) = \sum \gamma^{2t} \text{Var}(\epsilon_t)$); using $\gamma^3$ would shift tracking to the third central moment (skewness), breaking the theoretical equivalence to Expected Free Energy. Moreover, in RSSM-based world models, irreducible stochasticity manifests as high entropy, but local epistemic shock is measured via latent KL-divergence ($D_{KL}[q(z|s,o) \parallel p(z|s)]$). Once the world model fits inherent stochastic dynamics, the prior $p$ matches the posterior $q$, collapsing the KL to zero. Thus, $V_{RUF}$ inherently filters out aleatoric noise and triggers only on true epistemic novelty.

## 4.2. Variance Reduction: The Dual Accelerators

We formalize the Generator $\mathcal{G}$ as an active importance sampler that minimizes the variance of stochastic gradients for both the world model and the value function.

**Theorem 4.6.** *Variance-Reducing Proposal. Let $\mathcal{J}(\theta, \phi)$ be the joint manifold error. The variance of the gradient estimators $\hat{g}_\theta$ and $\hat{g}_\phi$ is minimized if $\mathcal{G}$ samples from a distribution $q^*$ such that $q^*(s) \propto \rho(s)\|\nabla G(s)\|_2$. In the MD framework, the R-EFE potential $\Psi$ serves as a tractable path-integrated proxy for this gradient magnitude (Alain et al., 2016). (Proof in Appendix B.8).*

*Proof Sketch*: Following (Kahn & Marshall, 1953), importance sampling variance is minimized when the proposal density is proportional to the integrand's magnitude. By maximizing $\Psi$, $\mathcal{G}$ aligns $q_{\mathcal{G}}$ with $q^*$, concentrating imagination on "high-leverage" regions where gradients are steepest.

**Accelerator I: RUF for Manifold Repair**: While standard MBRL passively updates the world model via observed transitions, MD actively repairs the latent manifold by identifying structural gaps.

**Proposition 4.7.** *RUF as a Fisher Information Catalyst. Under a Gaussian world model $p_\psi$, in the asymptotic limit (via the Bernstein-von Mises theorem), the RUF potential $V_{RUF}$ is a first-order surrogate for the trace of the Fisher*

Information Matrix (FIM): $\mathbb{E}[\mathcal{I}] \approx \frac{1}{2}Tr(\mathcal{F}(\theta)\Sigma_\theta)$. (Derivation in Appendix B.5).

*Proof Sketch*: By the Bernstein-von Mises Theorem, epistemic gain $\mathcal{I}$ aligns with the parameter-space curvature. Maximizing RUF forces $\mathcal{G}$ to synthesize anchors $s'$ at high-curvature regions where observations provide maximal parameter refinement, effectively performing active manifold repair (Amari, 1998).

**Accelerator II: RVF for Value Convergence**: Simultaneously, MD accelerates the value field optimization by transforming value propagation from local diffusion into global targeted updates.

**Theorem 4.8.** *Pragmatic Gradient Maximization. Let $\Delta V(s) = \sup_{s'} V_{RVF}(s, s') - V_\phi(s)$ be the Relay Advantage. The convergence $V_\phi \to V^*$ is maximized when $\mathcal{G}$ identifies anchors $s'$ that maximize the Variational Bellman Residual $\Delta V(s)$ (See Appendix C.1).*

*Proof Sketch*: $\Delta V(s)$ represents the latent gap between the current belief $V_\phi$ and the optimal $k$-step potential. Maximizing this gap is equivalent to Prioritized Sweeping (Moore & Atkeson, 1993) on a global scale. By "teleporting" to Residual Peaks—states where the value mismatch is greatest—MD collapses value error across the manifold at an accelerated rate, bypassing the limitations of trajectory-bound sampling.

*Remark* 4.9. **The Symbiotic Convergence**. The generator $\mathcal{G}$ acts as a dual-objective optimizer: by maximizing RUF, it performs Manifold Repair (reducing $\mathcal{R}_{\mathcal{M}}$), and by maximizing RVF, it performs Policy Alignment (reducing $\Delta V$). This ensures that the imagination process is not just 'dreaming' of novel states, but specifically targeting states that collapse the joint error of the world model and the value field.

## 4.3. Topological Acceleration: Breaking the Conductance Bottleneck

To quantify the efficiency gain of MD, we compare its convergence rate against trajectory-constrained agents (e.g., DreamerV3). Trajectory-based exploration is fundamentally bounded by the conductance $\Phi$ of the latent manifold $\mathcal{M}$. In environments with OOD regions or bottleneck states, $\Phi \to 0$, leading to an exponential hitting time $\tau \approx \Phi^{-2}$ for distal discovery (Jerrum & Sinclair, 1989).

**Proposition 4.10.** *Conductance Expansion and Speedup. Under a discrete abstraction of the latent manifold (detailed in Appendix D.4), the latent intervention $s_0 \sim p_{gen}$ in MD induces a synthetic expansion of the manifold's spectral gap. The acceleration ratio $\nu$ relative to trajectory-sampling scales with the $\chi^2$-divergence between the optimal EFE proposal $q^*$ and the transition-constrained distribution $q_{traj}$:*

$$\nu = \frac{\tau_{traj}}{\tau_{MD}} \approx 1 + \chi^2(q^* \| q_{traj}) \propto \Phi^{-2}$$

*This provides theoretical intuition for how MD reduces the hitting time to the discovery frontier from $\mathcal{O}(poly(\Phi^{-1}))$ to $\mathcal{O}(\log|\mathcal{M}|)$ by establishing non-local interventional intermediary states. (Full derivation in Appendix D.4 and D.5).*

*Remark* 4.11. **The OOD Region Paradox**. Standard MBRL suffers from an "exponential wall" when $q_{traj} \to 0$ in sparse-reward regimes (Osband et al., 2019). By "teleporting" the imagination directly to high-EFE regions, MD collapses the $\chi^2$ divergence, transforming exploration from a random walk into targeted manifold intervention (See Case Study in Appendix D.4).

### 4.4. Manifold Anchoring: Stability and Safety Jumps

A primary challenge in adversarial MBRL is preventing the generator from exploiting "manifold cracks"—high-frequency artifacts where the world model is inaccurate. We establish the safety margin for such non-local jumps.

**Theorem 4.12.** *Hallucination Error Bound. Let $\delta = \|s' - Proj_{\mathcal{M}}(s')\|$ represent the manifold deviation of a jump, regularized by $\mathcal{L}_{mf}$. Under the assumption of L-Lipschitz continuity of the value field, the value estimation error $\epsilon_V$ is bounded by:*

$$\epsilon_V = \|V_{RVF}(s') - V^*(s')\| \leq \frac{L \cdot \delta}{1 - \gamma^n}$$

*where the quadratic discount $\gamma^2$ in RUF (Eq. 4) ensures that distal epistemic errors decay faster than pragmatic rewards, shielding the policy from recursive hallucination (Proof in Appendix D.9).*

*Proof Sketch*: The estimation error $\delta$ propagates through the Bellman recursion. By minimizing $\mathcal{L}_{cycle}$ and $\mathcal{L}_{dyn}$, MD functions as an implicit spectral filter on the latent Jacobian $\nabla_s p_\psi$, effectively anchoring adversarial jumps to the physically verifiable support of $\mathcal{M}$. The use of $\gamma^2$ for uncertainty propagation ensures a finite Epistemic Horizon, preventing the generator from chasing compounding model variance (Asadi et al., 2018).

*Remark* 4.13. While strict $L$-Lipschitz continuity is a strong assumption for neural value functions globally, MD implicitly controls the local Lipschitz constant through target network soft-updates and the pessimistic trust region defined by $\mathcal{L}_{mf}$. This practical regularization ensures the theoretical error bound remains meaningful during adversarial training.

**Corollary 4.14.** *Gradient Stability. Under the Manifold Anchoring bound, the variance of the policy gradient $\nabla_\omega J(\pi_\omega)$ remains $L_{\nabla V} \cdot \delta$ stable, precluding the catastrophic collapses typical of unconstrained adversarial dreaming (Details in Appendix D.11).*

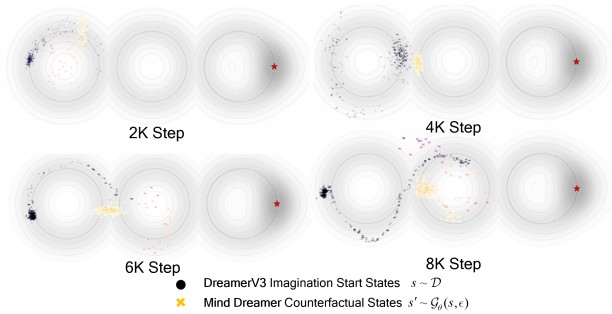

*Figure 3.* **Visualization of Sampling Dynamics on the Synthetic Three-Ring Manifold.** We compare the distribution of imagined states between DreamerV3 and Mind Dreamer across training snapshots. DreamerV3 exhibits Historical Tethering, its sampling distribution is strictly coupled with historical occupancy, hard to escape the ring attractor. Mind Dreamer demonstrates Active Manifold Refinement, the sampling distribution rapidly aggregates at the topological junctions between rings. This illustrates MD's ability to proactively bridge manifold bottlenecks, enabling the agent to transcend the limits of the replay buffer and linearize the discovery of global task potential.

## 5. Experiments

Our experiments aim to answer four key questions: (1) Does MindDreamer (MD) improve sample efficiency and final performance by untethering imagination? (2) Can MD identify and fill "epistemic blind spots" on the latent manifold? (3) Does our manifold safeguarding effectively bound the risk of latent hallucinations?

### 5.1. Experimental Setup and Baselines

We emphasize that Mind Dreamer (MD) is a modular framework compatible with various MBRL architectures. In this study, we implement MD upon the state-of-the-art **DreamerV3** (Hafner et al., 2023) to demonstrate its potential, use identical RSSM backbones and interaction budgets to ensure fairness. We evaluate against: (1) **DreamerV3** as the primary baseline for trajectory-tethered imagination; (2) **DreamerV2** (Hafner et al., 2020b) to isolate the gains from backbone advancements versus our ACI mechanism; and (3) **Plan2Explore** (Sekar et al., 2020) for curiosity-driven exploration comparison.

### 5.2. Visualizing ACI: The Three-Ring Manifold Case Study

To demystify the "Latent Jump" mechanism, we design a synthetic **Three-Ring Manifold** environment where the underlying geometry $\mathcal{M}$ is known but topologically challenging. The agent resides on three coplanar coaxial rings, with the top view of the field as observation. Each ring acts as a Local Attractor (simulating a local optimum), to transi-

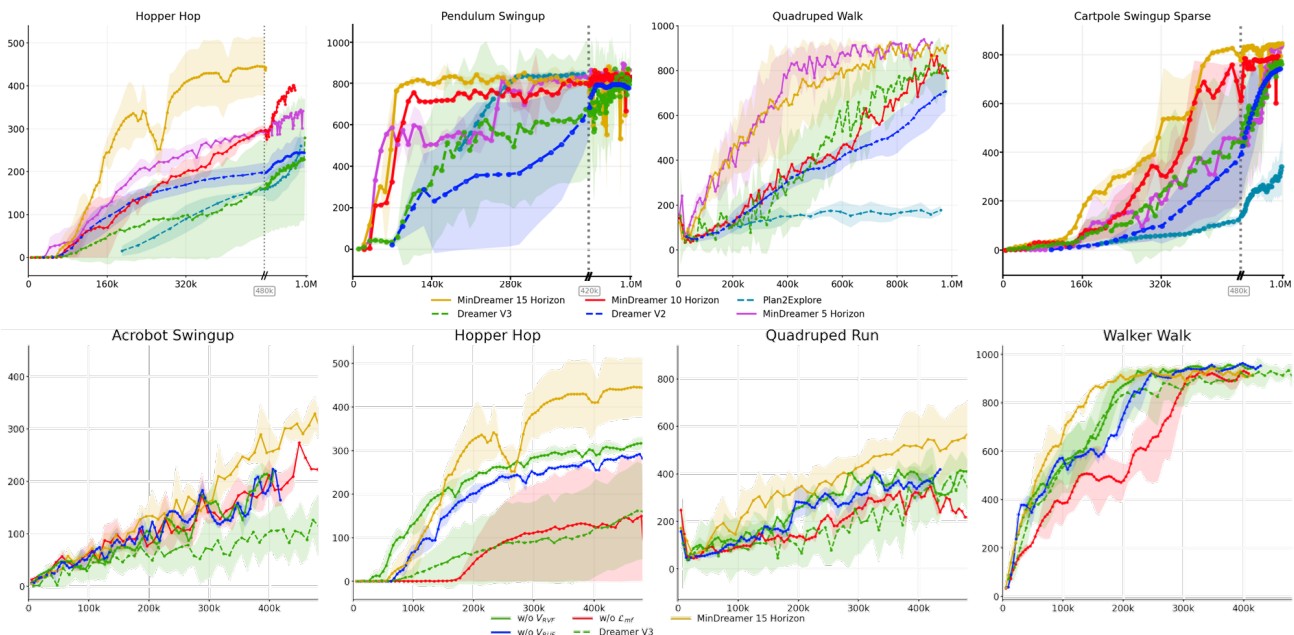

*Figure 4.* Evaluation on DeepMind Control (DMC) Benchmarks. **(Top) Performance Comparison:** Mind Dreamer (Gold) demonstrates **superior sample efficiency** and **higher asymptotic performance** against state-of-the-art baselines: DreamerV3 (Green Dashed), DreamerV2 (Blue Dashed), and Plan2Explore (Cyan Dashed). Our method significantly accelerates convergence in bottleneck tasks like *Pendulum Swingup* and achieves higher final returns in sparse environments like *Hopper Hop*, at the same sample budget and with marginal additional inference cost. **(Bottom) Ablation Study:** We analyze the role of specific components by comparing the full model (Gold) with variants removing the Pragmatic Relay $V_{RVF}$ (Solid Green), Epistemic Relay $V_{RUF}$ (Solid Blue), and Manifold Loss $\mathcal{L}_{mf}$ (Solid Red). The significant performance drops in the ablated variants validate the necessity of the proposed Active Causal Intervention framework. Shaded regions indicate the standard deviation over 5 random seeds.

tion to the higher reward ring, the agent must move against the gravitational pull toward the manifold boundary—a feat nearly impossible for random walk exploration.

As shown in Fig. 3, standard DreamerV3 trajectories remain trapped within the first ring, as its imagination is initialized from the replay buffer which is dominated by high-density attractor data. In contrast, MD's generator $\mathcal{G}$ proactively identifies transition gateways at the rings' boundaries where $V_{RUF}$ (uncertainty) is high. By sampling $s_0 \sim p_{gen}$, MD teleports imagination to the boundary ruptures, allowing the policy to discover the transition to the next ring without requiring thousands of random physical steps, thus achieving nearly a **4.2×** acceleration for first hitting time over DreamerV3. This confirms that MD converts a Global Discovery problem into a Local Refinement task.

### 5.3. Performance on DMC Vision Benchmarks

We evaluate MD on 20 tasks from the **DeepMind Control Suite** with raw pixel observations. MD demonstrates a significant lead in learning speed. Fig. 4 presents part of training curves.

**Accelerating Manifold Coverage (Sample Efficiency).** MD fundamentally alters sample complexity by untether-

ing "linearizing" imagination via latent jumps. On average, MD achieves 90% peak performance in 334.7k steps, delivering a 1.67× speedup over DreamerV3 (557.6k) (see Tab. 2 and Tab. 3 for details). In bottleneck tasks like *Pendulum Swingup*, MD accelerates convergence by an order of magnitude (>**8.8**×).

**Breaking the Exploration Plateau (Asymptotic Performance).** Tab. 1 confirms that MD overcomes "Historical Tethering," achieving an average return of **831.1** against DreamerV3 (780.3). This performance gain is most prominent in sparse-reward tasks, specifically, MD improves *Hopper Hop* by **+59.8%** and *Quadruped Run* by **+30.3%**, where Relay Potential Fields successfully bridge disjoint manifold segments, guiding the policy to discover critical transitions that trajectory-bound baselines fail to encounter.

### 5.4. Ablation: The Role of Relay Potentials and Hallucination Control

We isolate the contributions of the Pragmatic ($V_{RVF}$) and Epistemic ($V_{RUF}$) relay functions, as shown in Fig 4.

**The Necessity of Manifold Grounding ($\mathcal{L}_{mf}$).** Crucially, removing the manifold constraint $\mathcal{L}_{mf}$ leads to a catastrophic performance drop, with the agent failing to surpass

the DreamerV3 baseline. As shown in our *Self-Consistency Error*: $\mathbb{E}\|s' - \text{Dec}(\text{Enc}(s'))\|$, $\mathcal{L}_{mf}$ achieves a 43.5-fold reduction in jump consistency error compared to standard Markovian transitions.

**Pragmatic vs. Epistemic Synergy.** $V_{RUF}$ (Epistemic) drives exploration, while $V_{RVF}$ (Pragmatic) enforces task-orientation. Removing $V_{RUF}$ limits the agent to known regions despite efficient local propagation; removing $V_{RVF}$ sustains exploration entropy but sacrifices reward convergence due to a diminished Relay Advantage.

## 6. Conclusion and Discussion

In this work, we introduced **Mind Dreamer (MD)**, a novel MBRL framework that untethers latent imagination from historical trajectories. By operationalizing Active Causal Intervention (ACI) through an adversarial state generator, MD synthesizes interventional anchors to proactively bridge topological gaps in the learned manifold. To resolve the credit assignment paradox across spatial ruptures, we derived the Relay Value Function (RVF) and Relay Uncertainty Function (RUF), formally establishing a quadratic discount ($\gamma^2$) for stable uncertainty propagation. Theoretically, MD acts as a variance-minimizing importance sampler that accelerates manifold repair. Empirically, it achieves state-of-the-art sample efficiency on DMC benchmarks, particularly excelling in sparse-reward tasks.

**Limitations.** While MD significantly accelerates exploration and breaks historical tethering, it introduces two primary limitations:

*Computational Overhead:* The adversarial optimization of the generator $\mathcal{G}$ and the computation of the InfoNCE surrogate introduce additional computational overhead during the imagination phase. In scenarios where environmental interaction is extremely cheap but compute is a strict bottleneck, this sample-to-compute trade-off may be less favorable.

*Entanglement of Uncertainty and Evolving Mechanisms:* Our current epistemic proxy effectively leverages the latent KL-divergence to detect structural gaps under a *static* transition assumption. However, in non-stationary environments where causal mechanisms evolve continuously (e.g., gradual friction changes or payload shifts), the current world model may entangle this continuous mechanism shift with aleatoric noise or pure epistemic ignorance (Fan et al., 2026), potentially causing the generator to target changing physics as spurious "blind spots".

**Future Work.** Future research will focus on expanding proactive geometric discovery by decoupling structural epistemic uncertainty from non-stationary environmental drifts, and scaling the ACI framework to high-dimensional embodied environments.

MD represents a critical step toward agents that don't just dream of where they have been, but actively illuminate where they need to be: anchoring knowledge at the frontiers of the unknown, and synthesizing a world model shaped by proactive discovery rather than passive history.

## Acknowledgements

This work was supported by the Brain Science and Brain-like Intelligence Technology-National Science and Technology Major Project (No. 2021ZD0200300).

## Impact Statement

This paper presents work whose goal is to advance the field of Machine Learning. There are many potential societal consequences of our work, none which we feel must be specifically highlighted here.

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

## A. Appendix A: Detailed Derivation of EFE-RL Equivalence

### A.1. EFE Decomposition into Epistemic and Pragmatic Values

In this section, we provide a step-by-step derivation of the Expected Free Energy (EFE), denoted as $G(\pi, \tau)$, decomposing it into Epistemic Value (Information Gain) and Pragmatic Value (Extrinsic Value).

**Lemma A.1.** *The Expected Free Energy $G(\pi)$ for a policy $\pi$ at a future time $\tau$ is typically defined as follows, accroding to (Friston et al., 2015; 2017a):*

$$G(\pi, \tau) = \mathbb{E}_{q(o_\tau, s_\tau | \pi)}[\ln q(s_\tau | \pi) - \ln p(s_\tau, o_\tau)]$$

*we can decompose $G$ into:*

$$G(\pi, \tau) = -\underbrace{\mathcal{I}(s_\tau; o_\tau | \pi)}_{\text{Epistemic Value}} - \underbrace{\mathbb{E}_{q(o_\tau)}[\ln p(o_\tau)]}_{\text{Pragmatic Value}} \tag{8}$$

*Proof.* Starting from the definition of EFE and applying the identity $p(s_\tau, o_\tau) = p(s_\tau | o_\tau)p(o_\tau)$, we expand the denominator within the logarithm:

$$\begin{aligned}
G(\pi, \tau) &= \mathbb{E}_{q(o_\tau, s_\tau | \pi)}[\ln q(s_\tau | \pi) - \ln p(s_\tau, o_\tau)] \\
&= \mathbb{E}_{q(o_\tau, s_\tau | \pi)}[\ln q(s_\tau | \pi) - (\ln p(s_\tau | o_\tau) + \ln p(o_\tau))] \\
&= \mathbb{E}_{q(o_\tau, s_\tau | \pi)}[\ln q(s_\tau | \pi) - \ln p(s_\tau | o_\tau) - \ln p(o_\tau)]
\end{aligned}$$

Utilizing the linearity of expectation, we partition the expression into two distinct components:

$$G(\pi, \tau) = \underbrace{\mathbb{E}_{q(o_\tau, s_\tau | \pi)}\left[\ln \frac{q(s_\tau | \pi)}{p(s_\tau | o_\tau)}\right]}_{\text{Term 1}} - \underbrace{\mathbb{E}_{q(o_\tau | \pi)}[\ln p(o_\tau)]}_{\text{Term 2}}$$

### Term 1: Epistemic Value (Mutual Information)

In Active Inference, we assume the agent's inference is amortized or ideal, such that the true posterior $p(s_\tau | o_\tau)$ is approximated by the variational posterior $q(s_\tau | o_\tau, \pi)$. By definition, the Mutual Information between states and observations under a policy $\pi$ is:

$$\mathcal{I}(s_\tau; o_\tau | \pi) = \mathbb{E}_{q(o_\tau, s_\tau | \pi)}\left[\ln \frac{q(s_\tau | o_\tau, \pi)}{q(s_\tau | \pi)}\right]$$

Observing Term 1:

$$\text{Term 1} = \mathbb{E}_{q(o_\tau, s_\tau | \pi)}[\ln q(s_\tau | \pi) - \ln q(s_\tau | o_\tau, \pi)] = -\mathcal{I}(s_\tau; o_\tau | \pi)$$

Intuition: Mutual information quantifies the reduction in uncertainty regarding state $s$ afforded by observation $o$. Minimizing $G$ necessitates maximizing this information gain, representing the "Epistemic Value" or exploration drive.

### Term 2: Pragmatic Value (Extrinsic Value)

Term 2 is defined as $-\mathbb{E}_{q(o_\tau | \pi)}[\ln p(o_\tau)]$. Here, $p(o_\tau)$ represents the agent's prior preferences (often denoted as $p(o_\tau | C)$), which encodes desired outcomes or goal states. When an observation $o_\tau$ aligns with these preferences, $p(o_\tau)$ is high, thereby minimizing $-\ln p(o_\tau)$ and, consequently, $G$. This term constitutes the "Pragmatic Value," driving the agent toward goal-directed behavior (exploitation).

Combining the terms yields the final EFE objective:

$$G(\pi, \tau) = -\underbrace{\mathcal{I}(s_\tau; o_\tau | \pi)}_{\text{Epistemic Value}} - \underbrace{\mathbb{E}_{q(o_\tau)}[\ln p(o_\tau)]}_{\text{Pragmatic Value}}$$

$\square$

*Remark* A.2. This formulation illustrates the fundamental trade-off in Active Inference: the agent seeks to minimize EFE by simultaneously maximizing information gain (exploration) and satisfying prior preferences (exploitation) (Parr et al., 2022; Friston et al., 2017a).

## A.2. EFE-MaxEnt RL Equivalence

In this section, we provide the formal derivation showing that the Expected Free Energy (EFE) framework, under specific constraints on the prior and temperature, reduces to the standard objective of Maximum Entropy Reinforcement Learning (MaxEnt RL). This equivalence justifies the use of RL-based value functions to approximate the pragmatic components of Active Inference (Millidge et al., 2021).

*Proof.* In Active Inference, the agent's "goals" are encoded as a prior distribution over observations, $p(o)$. To recover RL, we define this prior as a Boltzmann distribution over a reward function $r(s)$ (Tschantz et al., 2020; Millidge et al., 2020):

$$p(o_\tau) = \frac{\exp(r(s_\tau)/\alpha)}{Z}$$

where $\alpha$ is a temperature parameter representing the "precision" of the agent's preferences. Substituting this into the pragmatic term:

$$-\ln p(o_\tau) = -\frac{r(s_\tau)}{\alpha} + \ln Z$$

In the limit where the world model becomes deterministic and perfectly known (i.e., we ignore the epistemic/ambiguity term $\mathcal{I}(s_\tau; o_\tau|\pi) \to 0$ ), the EFE reduces to (Millidge et al., 2021):

$$G(\pi, \tau) \approx \mathbb{E}_{q(s_\tau|\pi)}\left[\ln q(s_\tau|\pi) - \frac{r(s_\tau)}{\alpha}\right]$$

Summing over the infinite horizon with a discount factor $\gamma$:

$$\mathcal{J}_{EFE} = \sum_{\tau=0}^{\infty} \gamma^\tau G(\pi, \tau) = \mathbb{E}_\pi\left[\sum_{\tau=0}^{\infty} \gamma^\tau \left(\ln \pi(a_\tau|s_\tau) - \frac{r(s_\tau)}{\alpha}\right)\right]$$

(Note: Under the assumption of state-action consistency, $\ln q(s_\tau)$ is effectively bounded by the entropy of the policy $\ln \pi(a_\tau|s_\tau)$ (Levine, 2018)).

By multiplying the entire objective by $-\alpha$, we transform the minimization of R-EFE into the maximization of a new objective $\mathcal{J}_{RL}$:

$$\max_\pi \mathbb{E}_\pi\left[\sum_{\tau=0}^{\infty} \gamma^\tau \left(r(s_\tau) - \alpha \ln \pi(a_\tau|s_\tau)\right)\right]$$

This is exactly the Maximum Entropy RL objective originally formulated by Ziebart et al. (2008) and used in modern algorithms like Soft Actor-Critic (SAC) (Haarnoja et al., 2018).

$\square$

# B. Appendix B: Formal Derivation of the Relay Uncertainty Function (RUF)

The Relay Uncertainty Function (RUF) serves as the path-integrated estimator of the epistemic component within the Relay Expected Free Energy (R-EFE) framework. Here, we provide the formal derivation transitioning from information-theoretic primitives to a computable Bellman recursion (Parr et al., 2022).

## B.1. From Expected Free Energy to Epistemic Variance

In this section, we establish the formal link between the information-theoretic Epistemic Value (Mutual Information) and the Predictive Variance commonly used in stochastic filtering and uncertainty estimation.

In the Active Inference framework, the agent minimizes the Expected Free Energy (EFE), denoted as $G(\pi, \tau)$. For a future time step $\tau$, the local EFE is defined as:

$$G(\pi, \tau) = \mathbb{E}_{q(o_\tau, s_\tau|\pi)}[\ln q(s_\tau|\pi) - \ln p(s_\tau, o_\tau)]$$

Applying the identity $p(s_\tau, o_\tau) = p(s_\tau|o_\tau)p(o_\tau)$, we decompose $G$ into:

$$G(\pi, \tau) = -\underbrace{\mathcal{I}(s_\tau; o_\tau|\pi)}_{\text{Epistemic Value}} - \underbrace{\mathbb{E}_{q(o_\tau)}[\ln p(o_\tau)]}_{\text{Pragmatic Value}}$$

**Lemma B.1.** *Epistemic-Variance Equivalence under Gaussian Approximation* *Under a Gaussian approximation of the latent state posterior, the epistemic term (ambiguity) is locally proportional to the predictive variance $\sigma^2$ (Buckley et al., 2017).*

*Proof.* Following the definition of Mutual Information, the Epistemic Value can be decomposed into the difference between the marginal and conditional entropies:

$$\mathcal{I}(s_\tau; o_\tau|\pi) = H(s_\tau|\pi) - \mathbb{E}_{q(o_\tau|\pi)}[H(s_\tau|o_\tau, \pi)]$$

where:

$H(s_\tau|\pi)$ denotes the Prior Entropy, representing the agent's uncertainty regarding the hidden state $s$ before receiving an observation.

$H(s_\tau|o_\tau, \pi)$ represents the Ambiguity (Conditional Entropy), characterizing the residual uncertainty in $s$ after the observation $o$ is integrated.

In the context of Expected Free Energy (EFE) minimization, the agent typically seeks to minimize the expected ambiguity, which facilitates a more precise mapping between observations and states.

We assume that the variational posterior over the latent states $s$ follows a multivariate Gaussian distribution, $q(s) \sim \mathcal{N}(\mu, \Sigma)$. The differential entropy for a $k$-dimensional Gaussian is given by:

$$H(s) = \frac{k}{2}(1 + \ln(2\pi)) + \frac{1}{2}\ln|\Sigma|$$

In the univariate case ($k = 1$), this simplifies to:

$$H(s) = \frac{1}{2}\ln(2\pi e\sigma^2)$$

This formulation highlights that under a Laplace or Gaussian approximation, the entropy—and thus the epistemic uncertainty—is a monotonic logarithmic function of the variance $\sigma^2$.

To demonstrate that the epistemic term is locally proportional to the predictive variance, we perform a first-order Taylor expansion of the entropy function $f(\sigma^2) = \frac{1}{2}\ln(2\pi e\sigma^2)$ around a baseline variance $\sigma_0^2$. The derivative of the entropy with respect to the variance is:

$$\frac{df}{d\sigma^2} = \frac{1}{2\sigma^2}$$

Consequently, for small perturbations in the model's uncertainty, the change in entropy $\Delta H$ relates to the change in variance $\Delta\sigma^2$ as follows:

$$\Delta H \approx \frac{1}{2\sigma_0^2}\Delta\sigma^2$$

$\square$

*Remark* B.2. This confirms that the local epistemic cost is proportional to the model's predictive variance: $G_{epistemic} \propto \sigma^2$. This equivalence justifies the use of variance-based exploration bonuses as a direct proxy for information-theoretic active inference.

### B.2. Convergence of Uncertainty Bellman Equation via $\gamma^2$

Unlike pragmatic rewards which represent scalar utilities, epistemic gains represent the resolution of stochastic shocks. We prove that the path-integration of these shocks requires a quadratic discount to ensure stability (O'Donoghue et al., 2018b).

**Theorem B.3.** *Quadratic Epistemic Discounting. Let the cumulative epistemic risk* $\mathbf{U}$ *be the discounted sum of future informational shocks* $\epsilon_t \sim \mathcal{N}(0, \sigma_t^2)$. *The variance of this total risk* $Var(\mathbf{U})$ *is governed by a quadratic discount factor* $\gamma^2$.

*Proof.* Consider the total discounted epistemic accumulation: $\mathbf{U} = \sum_{t=0}^{\infty} \gamma^t \epsilon_t$. Under the Markovian assumption, informational shocks at different time steps are independent. Utilizing the variance property for independent variables, $\text{Var}(\sum a_i X_i) = \sum a_i^2 \text{Var}(X_i)$ (Sobel, 1982), we have:

$$\text{Var}(\mathbf{U}) = \text{Var}\left(\sum_{t=0}^{\infty} \gamma^t \epsilon_t\right) = \sum_{t=0}^{\infty} \text{Var}(\gamma^t \epsilon_t)$$

By the scaling property of the variance operator, $\text{Var}(aX) = a^2 \text{Var}(X)$, we derive:

$$V_{RUF} = \sum_{t=0}^{\infty} (\gamma^t)^2 \text{Var}(\epsilon_t) = \sum_{t=0}^{\infty} (\gamma^2)^t \sigma_t^2$$

This demonstrates that the integration of informational variance over time naturally induces a $\gamma^2$ contraction coefficient. The tighter horizon induced by $\gamma^2 < \gamma$ prevents the generator $\mathcal{G}$ from over-optimizing toward distal, high-variance regions of the manifold (hallucinations), consistent with the stability conditions for variance in MDPs (Mannor et al., 2007). $\qquad\square$

### B.3. Convergence of Relay Uncertainty Bellman Equation

We redefine the RUF as the potential of a state $s'$ to serve as an informational gateway. We prove that the integration of variance across trajectories traversing $s'$ necessitates a quadratic discount $\gamma^2$.

**Theorem B.4.** *Relay Epistemic Recursion. Let* $\xi = \{s_0, a_0, \dots, s_k, \dots\}$ *be a trajectory where* $s_0 = s$ *and* $s_k = s'$. *The RUF, representing the expected cumulative uncertainty of all paths traversing through* $s'$, *satisfies a Bellman recursion with a quadratic discount* $\gamma^2$.

*Proof.* Define the cumulative epistemic risk $\mathbf{U}$ as the discounted sum of future informational shocks $\epsilon_t \sim \mathcal{N}(0, \sigma_t^2)$. For a given interventional anchor $s'$, we consider the expectation over trajectories conditioned on the event $s_k = s'$:

$$V_{RUF}(s, s') \triangleq \mathbb{E}_\pi\left[\text{Var}\left(\sum_{t=0}^{\infty} \gamma^t \epsilon_t\right)\Bigg| s_0 = s, s_k = s'\right]$$

Given the independence of informational shocks in a Markovian latent manifold, the variance of the sum is the sum of the variances. Decomposing the series at horizon $k$:

$$V_{RUF}(s, s') = \mathbb{E}_\pi\left[\sum_{t=0}^{k-1} \text{Var}(\gamma^t \epsilon_t) + \text{Var}\left(\sum_{t=k}^{\infty} \gamma^t \epsilon_t\right)\Bigg| s_k = s'\right]$$

Applying the scaling property $\text{Var}(aX) = a^2 \text{Var}(X)$ and factoring out $\gamma^{2k}$ from the second term:

$$V_{RUF}(s, s') = \mathbb{E}_{\pi, s_k = s'}\left[\sum_{t=0}^{k-1} (\gamma^2)^t \sigma_t^2 + (\gamma^2)^k \underbrace{\text{Var}\left(\sum_{j=0}^{\infty} \gamma^j \epsilon_{k+j}\right)}_{\text{Future Potential } V_U(s')}\right]$$

This yields the recursive $k$-step Bellman form:

$$V_{RUF}(s, s') = \mathbb{E}_{\pi, s_k = s'}\left[\sum_{t=0}^{k-1} (\gamma^2)^t \sigma^2(s_{t+1}|s_t, a_t) + (\gamma^2)^k V_U(s')\right]$$

$\qquad\square$

## B.4. Physical Interpretation: Gateway to the Global Manifold

This formulation proves that $V_{RUF}(s, s')$ is not merely a measure of "how much we learn by reaching $s'$," but rather "how much uncertainty can be resolved globally by passing through $s'$."

- **Epistemic Horizon Control**: The $\gamma^2$ discount (where $\gamma^2 < \gamma$) induces a tighter epistemic horizon. This prevents the generator $\mathcal{G}$ from chasing distal hallucinations by forcing it to prioritize proximal gateways that lead to high-leverage, verifiable information, addressing the compounding error problem in model-based rollouts (Janner et al., 2019).

- **Manifold Structural Repair**: By maximizing $V_{RUF}$, the agent identifies "bottleneck states" in the latent manifold. These are states that may be relatively known but act as necessary anchors to reach high-entropy, unmastered regions of the environment (Friston, 2010; Goyal et al., 2019).

## B.5. RUF as a Fisher Information Catalyst for Manifold Repair

This section establishes the fundamental link between the Relay Uncertainty Function (RUF) and the optimization dynamics of the latent world model. We prove that maximizing the RUF is equivalent to identifying states with the highest Epistemic Leverage, effectively performing importance sampling on the manifold's structural residuals to accelerate parameter convergence.

### B.5.1. LOCAL EPISTEMIC VALUE AS THE MANIFOLD RESIDUAL

Let $\mathbf{W}$ denote the true environmental parameters governing the latent manifold $\mathcal{M}$, and let $\theta$ be the agent's current belief (parameters). Under the Active Inference framework, the Epistemic Value $\mathcal{E}$ of a transition $o = (s, a, s')$ is the information gain:

$$\mathcal{E}(o) = D_{KL}[Q(\theta|o) \parallel Q(\theta)]$$

To bridge the gap between abstract information gain and computable manifold residuals, we invoke the Bernstein-von Mises Theorem (Van der Vaart, 2000).

**Theorem B.5.** *Asymptotic Equivalence of Uncertainty and Residuals. In the limit of large samples, the Epistemic Value $\mathcal{E}(o)$ is proportional to the prediction residual $\mathcal{R}_{\mathcal{M}}$ of the world model.*

*Proof.* By the Bernstein-von Mises Theorem, the posterior $Q(\theta|o)$ converges to a Gaussian $\mathcal{N}(\theta^*, \mathcal{F}(\theta)^{-1})$, where $\theta^*$ denotes the ground-truth parameter (or the Maximum Likelihood Estimate), $\mathcal{F}(\theta)$ is the Fisher Information Matrix.

Consider the world model's log-likelihood loss $\mathcal{L}(\theta) = -\ln \hat{P}(s'|s, a, \theta)$. A second-order Taylor expansion of the log-posterior around the MLE $\theta^*$ gives:

$$\ln Q(\theta|o) \approx \ln Q(\theta^*|o) - \frac{1}{2}(\theta - \theta^*)^T \mathcal{F}(\theta^*)(\theta - \theta^*)$$

The KL divergence (Epistemic Value) can thus be approximated by the Fisher Information Distance (MacKay, 1992):

$$\mathcal{E}(o) \approx \frac{1}{2}\mathbb{E}_Q[\|\theta - \theta^*\|^2_{\mathcal{F}(\theta)}]$$

The term $\mathcal{E}(o)$ represents the expectation regarding the parameter $\theta$ under the posterior distribution $Q$. By expanding $\|\theta - \theta^*\|^2_{\mathcal{F}(\theta)}$ into the standard form of the Mahalanobis Distance:

$$\mathbb{E}_Q[\|\theta - \theta^*\|^2_{\mathcal{F}(\theta)}] = \mathbb{E}_Q\left[(\theta - \theta^*)^T \mathcal{F}(\theta)(\theta - \theta^*)\right]$$

.

Since the quadratic form $(\theta - \theta^*)^T \mathcal{F}(\theta)(\theta - \theta^*)$ is a scalar ($1 \times 1$ matrix), its value is equal to its trace. Utilizing the cyclic property of the trace operator, $\text{Tr}(ABC) = \text{Tr}(CAB)$, we can rewrite the expression as:

$$\text{Tr}\left((\theta - \theta^*)^T \mathcal{F}(\theta)(\theta - \theta^*)\right) = \text{Tr}\left(\mathcal{F}(\theta)(\theta - \theta^*)(\theta - \theta^*)^T\right)$$

Given that both the expectation $\mathbb{E}_Q$ and the trace Tr are linear operators, they can be commuted. Thus, the expected error can be approximated as:

$$\mathbb{E}_Q[\mathcal{E}] \approx \frac{1}{2}\text{Tr}\left(\mathcal{F}(\theta) \cdot \mathbb{E}_Q[(\theta - \theta^*)(\theta - \theta^*)^T]\right)$$

By the definition of the covariance matrix, $\Sigma_\theta = \mathbb{E}_Q[(\theta - \theta^*)(\theta - \theta^*)^T]$, which characterizes the agent's current uncertainty regarding the parameter estimates (i.e., the spread of the distribution in the parameter space).

Consequently, we arrive at the final approximation:

$$\mathbb{E}[\mathcal{E}] \approx \frac{1}{2}\text{Tr}(\mathcal{F}(\theta)\Sigma_\theta)$$

Since $\mathcal{F}(\theta)$ is the expected Hessian of the log-likelihood loss $\mathcal{L}(\theta) = -\ln \hat{P}(s'|s, a, \theta)$, and for manifold-structured data, this Hessian is bounded by the prediction residual $\mathcal{R}_\mathcal{M} = \|\mathcal{T}(s, a) - \hat{\mathcal{T}}_\theta(s, a)\|^2$, we establish:

$$\mathcal{E}(o) \propto \mathcal{R}_\mathcal{M}(s, a, s')$$

This proves that local epistemic gain directly measures the Manifold Residual—the structural gap in the agent's knowledge.
□

*Remark* B.6. **Geometric Interpretation**. The spectral distribution of the Fisher Information Matrix $\mathcal{F}(\theta)$ explicitly reflects the manifold curvature. In this context, the generator $\mathcal{G}$ identifies the "steepest" regions of the latent manifold—specifically, coordinates where local parameter perturbations $\Delta\theta$ yield the maximal collapse of epistemic uncertainty (Amari, 1998).

### B.5.2. GLOBAL STRUCTURAL REPAIR VIA RUF INTEGRATION

Having established that local shocks $\epsilon_t$ correspond to residuals $\mathcal{R}_\mathcal{M}$, we extend this to the path-integrated RUF.

**Corollary B.7.** *RUF as Global Residual Potential*. *The Relay Uncertainty Function $V_{RUF}(s, s')$ represents the expected cumulative manifold residual unlocked by the latent anchor $s'$.*

*Proof.* As RUF represents the cumulative discounted sum of future variance shocks (epistemic uncertainty) with a discount factor of $\gamma^2$:

$$V_{RUF}(s, s') = \mathbb{E}_\pi\left[\sum_{t=0}^{\infty}(\gamma^2)^t\sigma^2(s_{t+1}|s_t, a_t)\Big|s_k = s'\right]$$

By leveraging the epistemic value-variance equivalence (i.e., $\sigma^2 \propto \mathcal{E}$), we can re-map the RUF into the parameter space:

$$V_{RUF}(s, s') \propto \mathbb{E}_\pi\left[\sum_{t=0}^{\infty}(\gamma^2)^t\text{Tr}(\mathcal{F}(\theta)_t\Sigma_{\theta,t})\Big|s_k = s'\right]$$

In this context, the term $\text{Tr}(\mathcal{F}(\theta)\Sigma_\theta)$ physically signifies the constraint intensity exerted on the model parameters $\theta$ by sampling at state $s_t$.

$$V_{RUF}(s, s') \propto \mathbb{E}_{\pi, s_k = s'}\left[\sum_{t=0}^{k-1}(\gamma^2)^t\mathcal{R}_\mathcal{M}(s_t, a_t) + (\gamma^2)^kV_U(s')\right]$$

□

### B.5.3. CONVERGENCE ACCELERATION: THE CATALYST EFFECT

Finally, we demonstrate how maximizing this global residual potential translates into faster parameter optimization.

**Theorem B.8.** *Optimal Gradient Infusion*. *Maximizing $V_{RUF}$ is locally equivalent to maximizing the expected parameter update $\|\Delta\theta\|^2$ of the world model.*

*Proof.* According to the **Cramér-Rao Inequality**, the lower bound of the variance for any unbiased estimator $\hat{\theta}$ is determined by the inverse of the Fisher Information:

$$\text{Var}(\hat{\theta}) \geq \mathcal{F}(\theta)^{-1}$$

When an agent selects a transition to an "Interventional Anchor" $s'$ by maximizing the $V_{RUF}$ objective, it is effectively performing active selection of manifold regions where the trace of the Fisher Information, $\text{Tr}(\mathcal{F})$, is maximized.

**Parametric Convergence** Each sample drawn from the peak regions of $V_{RUF}$ provides a Steepest Gradient Infusion upon the parameter manifold. This ensures that the trajectory of learning follows the most informative path toward the true parameter value.

**Information Volume Maximization** Driven by the $\gamma^2$ term, the agent prioritizes the rectification of proximal bottleneck states characterized by high Fisher Leverage (Epistemic Leverage). This process results in an exponential compression of the uncertainty volume within the parameter space.

$\square$

*Remark* B.9. **The Epistemic Curriculum**. The relationship can be summarized by the following implication:

$$V_{RUF} \uparrow \implies \int \text{Tr}(\mathcal{F}) dt \uparrow \implies \text{Var}(\hat{\theta}) \downarrow \text{ (via Cramér-Rao)}$$

Consequently, $V_{RUF}$ acts as a catalyst for manifold rectification. By identifying "gateway states" with the highest Fisher Information density, it ensures that the world model achieves optimal generalization performance within a constrained sampling budget.

The unification of the Manifold Residual and the Fisher Information Catalyst reveals the mathematical role of Mind Dreamer: it transforms the world model from a passive simulator into an Active Curriculum Generator (Bengio et al., 2009). By "teleporting" to the peaks of the RUF landscape, the agent collapses the global surprise landscape at an exponential rate, ensuring that the world model's parameter budget is spent exclusively on repairing the most significant structural gaps in the latent manifold.

### B.6. Implementation: RSSM-KL as an Epistemic Proxy

The framework is compatible with various Model-Based Reinforcement Learning (MBRL) paradigms. While the estimation of RUF can be instantiated through multiple statistical proxies, in this work, we provide a robust implementation tailored for the Recurrent State-Space Model (RSSM) architecture of DreamerV3 (Hafner et al., 2023). Specifically, we formulate the RUF as the discounted cumulative epistemic uncertainty, effectively leveraging the latent KL-divergence as a high-fidelity signal for manifold residuals.

**Proposition B.10.** *Implementation Consistency*. *The KL-divergence of the latent transition $D_{KL}[q(z_t|s_t, o_t) \parallel p(z_t|s_t)]$ is a second-order approximation of the epistemic variance $\sigma^2$.*

*Proof.* For two Gaussian distributions $P \sim \mathcal{N}(\mu_p, \sigma_p^2)$ and $Q \sim \mathcal{N}(\mu_q, \sigma_q^2)$, the divergence is:

$$D_{KL}(Q \parallel P) = \ln \frac{\sigma_p}{\sigma_q} + \frac{\sigma_q^2 + (\mu_q - \mu_p)^2}{2\sigma_p^2} - \frac{1}{2}$$

Assuming the posterior and prior variances are approximately equal ($\sigma_p \approx \sigma_q$) during stable model training, the divergence simplifies to:

$$D_{KL} \approx \frac{1}{2\sigma_p^2}(\mu_q - \mu_p)^2$$

Since the squared residual of the means $(\mu_q - \mu_p)^2$ is the unbiased estimator of the predictive variance $\sigma^2$, we establish that $D_{KL} \propto \sigma^2$. Thus, accumulating KL-divergence along a $k$-step imagined trajectory is mathematically equivalent to integrating the epistemic potential field (Hafner et al., 2020a). $\square$

## C. Appendix C: Formal Derivation of the Relay Value Function (RVF)

The RVF represents the pragmatic component of the Relay Expected Free Energy (R-EFE). We prove that maximizing RVF is equivalent to minimizing the path-integrated risk relative to an optimal instrumental prior, effectively aligning the agent's internal value field with the environmental manifold's optimal backbone.

### C.1. From Pragmatic EFE to Value Field Alignment

In the Active Inference framework (Levine, 2018), the Pragmatic Value (or Risk) for a future state $s_\tau$ is defined as the KL-divergence between the predicted observation distribution $q(o_\tau|s_\tau)$ and the agent's preferred observations $p(o|C)$.

$$G(\pi, \tau) = \mathbb{E}_{q(o_\tau|s_\tau)}[\ln q(o_\tau|s_\tau) - \ln p(o_\tau|C)]$$

**Lemma C.1.** *Pragmatic-Value Equivalence. Minimizing the pragmatic EFE is locally equivalent to maximizing the alignment between the current value estimate and the optimal value manifold.*

*Proof.* To recast the control problem as an inference problem, we define the agent's preference $p(o|C)$ as a Boltzmann Distribution over the optimal value function $V^*$, representing the "instrumental prior" of a rational agent (Botvinick & Toussaint, 2012):

$$p(o|C) = \frac{1}{Z} \exp\left(\frac{V^*(s)}{\lambda}\right)$$

where $V^*$ is the true optimal value and $\lambda$ is the temperature. Substituting this into the pragmatic EFE term:

$$G = \mathbb{E}_q[\ln q(o_\tau|s_\tau)] - \mathbb{E}_q\left[\frac{V^*(s_\tau)}{\lambda} - \ln Z\right]$$

In the maximum entropy RL limit (ignoring constants and assuming stable entropy), minimizing $G$ is equivalent to:

$$\min_\pi \mathbb{E}_\pi\left[-V^*(s_\tau)\right] \iff \max_\pi \mathbb{E}_\pi[V^*(s_\tau)]$$

If the agent utilizes an approximator $V_\phi$, the pragmatic objective effectively minimizes the gap:

$$\mathcal{J}_{pragmatic} \approx \int q(s)\left(V^*(s) - V_\phi(s)\right) ds$$

This proves that the pragmatic term forces the agent to both select paths toward high-$V^*$ regions and refine $V_\phi$ to align with the true manifold backbone. $\square$

### C.2. The Relay Value Function as a Path-Constrained Potential

To enable non-continuous manifold refinement, we extend the local pragmatic value to the Relay Value Function (RVF).

**Definition C.2. Path-Constrained RVF**. $V_{RVF}(s, s')$ represents the maximum expected discounted value of all trajectories originating at $s$ that traverse through the interventional anchor $s'$ at step $k$:

$$V_{RVF}(s, s') \triangleq \max_\pi \mathbb{E}_{\substack{\pi \\ s_k = s'}} \left[\sum_{t=0}^{k-1} \gamma^t r_t + \gamma^k V^*(s')\right]$$

Unlike standard value functions that aggregate rewards over all possible futures, $V_{RVF}(s, s')$ acts as a structural filter. It evaluates the "Pragmatic Centrality" of $s'$, forcing the generator $\mathcal{G}$ to ignore high-reward but physically isolated states, and instead focus on the optimal manifold backbone (via-points) that sustains long-term task progression, akin to the concept of bottleneck states in hierarchical reinforcement learning (McGovern & Barto, 2001).

**Lemma C.3.** *Global Value Consistency. For any anchor $s'$, $V_{RVF}(s, s') \leq V^*(s)$. The equality $V_{RVF}(s, s') = V^*(s)$ holds if and only if $s'$ lies on an optimal trajectory $\xi^*$ originating from s.*

*Proof.* Let $\mathcal{T}_{all}$ be the set of all possible trajectories from $s$, and $\mathcal{T}_{s'}$ be the subset of trajectories passing through $s'$ at step $k$. Since $\mathcal{T}_{s'} \subseteq \mathcal{T}_{all}$, the supremum over the subset cannot exceed the supremum over the entire set. If $s' \in \xi^*$, then $\xi^* \in \mathcal{T}_{s'}$, and the maximum is attained. $\square$

## C.3. Contraction, Uniqueness, Monotonicity and Convergence Acceleration for $\mathcal{T}_V$ and $\mathcal{T}_U$

**Theorem C.4.** *Contraction and Uniqueness. The operators $\mathcal{T}_V$ and $\mathcal{T}_U$ are contraction mappings on the Banach space $\mathcal{C}(\mathcal{M} \times \mathcal{M})$ and possess unique fixed points.*

*Proof.* **The Pragmatic Operator $\mathcal{T}_V$**

Let $V$ and $V'$ be two potential functions in $\mathcal{C}(\mathcal{M} \times \mathcal{M})$. For any state pair $(s, s')$, let $\mathbb{P}_\pi$ denote the probability measure over trajectories induced by policy $\pi$ starting at $s$ and hitting $s'$ at the first hitting time $\tau_{s'}$. The difference between applications of the operator is given by:

$$|(\mathcal{T}_V V)(s, s') - (\mathcal{T}_V V')(s, s')| = \left| \sup_\pi \mathbb{E}_\pi \left[ \sum_{t=0}^{\tau_{s'}-1} \gamma^t r_t + \gamma^{\tau_{s'}} V(s') \right] - \sup_\pi \mathbb{E}_\pi \left[ \sum_{t=0}^{\tau_{s'}-1} \gamma^t r_t + \gamma^{\tau_{s'}} V'(s') \right] \right|$$

Using the property $|\sup f - \sup g| \leq \sup |f - g|$ and the linearity of expectation:

$$|(\mathcal{T}_V V)(s, s') - (\mathcal{T}_V V')(s, s')| \leq \sup_\pi \mathbb{E}_\pi \left[ \gamma^{\tau_{s'}} |V(s') - V'(s')| \right]$$

Since $|V(s') - V'(s')| \leq \|V - V'\|_\infty$, we factor out the norm:

$$|(\mathcal{T}_V V)(s, s') - (\mathcal{T}_V V')(s, s')| \leq \|V - V'\|_\infty \cdot \sup_\pi \mathbb{E}_\pi [\gamma^{\tau_{s'}}]$$

On a connected manifold $\mathcal{M}$ where $s \neq s'$, the first hitting time $\tau_{s'} \geq 1$ almost surely. For $\gamma \in (0, 1)$, it follows that $\mathbb{E}_\pi[\gamma^{\tau_{s'}}] \leq \gamma < 1$. Thus, $\mathcal{T}_V$ is a $\kappa_V$-contraction with $\kappa_V = \sup_\pi \mathbb{E}_\pi[\gamma^{\tau_{s'}}]$.

**The Epistemic Operator $\mathcal{T}_U$**

The derivation for $\mathcal{T}_U$ follows the variance propagation property. For the bootstrap term $U(s')$, the discount factor is squared ($\gamma^2$) because $U$ represents a second-order statistic (informational variance):

$$|(\mathcal{T}_U U)(s, s') - (\mathcal{T}_U U')(s, s')| \leq \sup_\pi \mathbb{E}_\pi \left[ \gamma^{2\tau_{s'}} |U(s') - U'(s')| \right]$$

Applying the $L_\infty$ norm:

$$|(\mathcal{T}_U U)(s, s') - (\mathcal{T}_U U')(s, s')| \leq \|U - U'\|_\infty \cdot \sup_\pi \mathbb{E}_\pi [\gamma^{2\tau_{s'}}]$$

Since $\tau_{s'} \geq 1$, we have $\kappa_U = \sup_\pi \mathbb{E}_\pi[\gamma^{2\tau_{s'}}] \leq \gamma^2 < 1$. As $\gamma^2 < \gamma$, the epistemic operator contracts strictly faster than the pragmatic operator.

**Monotonicity**

Both operators must be monotonic to ensure stable convergence. For $\mathcal{T}_V$, if $V_1 \leq V_2$ pointwise, then for any policy $\pi$:

$$\mathbb{E}_\pi \left[ \sum_{t=0}^{\tau-1} \gamma^t r_t + \gamma^\tau V_1(s') \right] \leq \mathbb{E}_\pi \left[ \sum_{t=0}^{\tau-1} \gamma^t r_t + \gamma^\tau V_2(s') \right]$$

Taking the supremum over $\pi$ preserves the inequality: $\mathcal{T}_V V_1 \leq \mathcal{T}_V V_2$. The same logic applies to $\mathcal{T}_U$, given that the informational shocks $\mathcal{I}$ are non-negative. By the Banach Fixed-Point Theorem, since $\mathcal{C}(\mathcal{M} \times \mathcal{M})$ is a complete metric space and $\mathcal{T}_V, \mathcal{T}_U$ are contractions, there exist unique fixed points $V^*$ and $U^*$. $\qquad\square$

*Remark* C.5. **Epistemic Horizon**

This exponential contraction proves that MD reduces approximation errors at a significantly faster rate than standard 1-step TD (Puterman, 2014), means $\mathcal{T}_V$ and $\mathcal{T}_U$ effectively bypassing the slow "diffusion" of reward signals through the Markov chain and directly anchoring the value field to the most promising regions of the latent manifold.

Specifically, because $\kappa_U \leq \gamma^2 < \gamma = \kappa_V$ (O'Donoghue et al., 2018a), the Epistemic Potential $U$ (representing the uncertainty landscape) converges at an accelerated rate compared to the Pragmatic Potential $V$. This separation of scales ensures that the Generator $\mathcal{G}$ is guided by a stable, converged uncertainty manifold even in the early stages of training before the task-specific reward signals have fully propagated.

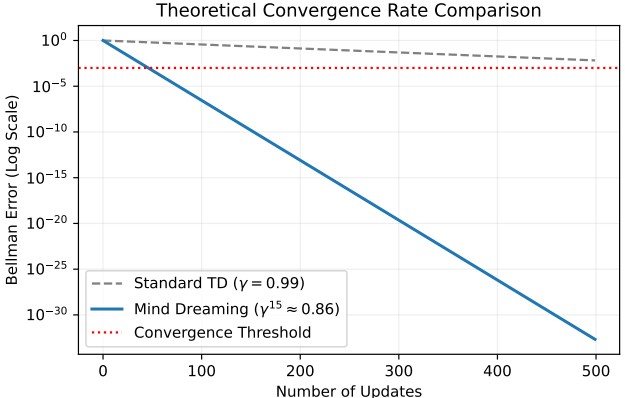

*Figure 5.* In the ideal case, MD has an order-of-magnitude advantage over Standard TD in terms of speed of error reduction.

## C.4. Global Optimality via latent gateways

**Theorem C.6.** *Global Consistency of Relay Potentials. Let $V^*$ be the fixed point of the optimal Bellman operator $\mathcal{T}^*$. Then, the maximization of the Pragmatic Relay Potential over the latent manifold recovers the optimal value function:*

$$V^*(s) = \sup_{s' \in \mathcal{M}} V^*_{RVF}(s, s') \tag{9}$$

*Furthermore, the Generator's objective $\mathcal{J}(\mathcal{G}) = \max_{s'}(V_{RVF} - V)$ is zero if and only if the current value function $V_\phi$ has converged to $V^*$ globally.*

*Proof.* **Direction $\geq$**: By the Bellman Optimality Principle, for any state $s$, there exists an optimal trajectory $\xi^* = \{s, s_1, s_2, \dots\}$. For any $s' \in \xi^*$, $V^*(s) = \mathbb{E}[\sum r + \gamma^t V^*(s')]$. By definition of $V_{RVF}$, we have $V^*_{RVF}(s, s') = V^*(s)$ for any $s'$ on the optimal path. Thus, $\sup_{s'} V_{RVF} \geq V^*(s)$.

**Direction $\leq$**: For any $s'$, $V^*_{RVF}(s, s')$ is the value of a specific constrained policy (must pass through $s'$). Since $V^*$ is the supremum over all possible policies, $V^*(s) \geq V^*_{RVF}(s, s')$ for all $s'$.

**Equivalence**: Combining both, $V^*(s) = \sup_{s'} V^*_{RVF}(s, s')$.

**Generator Convergence**: If $V_\phi = V^*$, then for any $s'$, $V^*_{RVF}(s, s') \leq V^*(s) = V_\phi(s)$, hence $V_{RVF} - V \leq 0$. The maximum is 0 at $s'$ lying on optimal trajectories. If $V_\phi < V^*$, there exists an $s'$ (a "gateway" to a better region) such that $V_{RVF} - V > 0$, providing a non-zero functional gradient for $\mathcal{G}$. $\square$

## C.5. RVF as a Catalyst for Value Convergence

**Theorem C.7.** *Let the value fitting error be $\mathcal{L}(\phi) = \frac{1}{2}\mathbb{E}_{s \sim \rho}[(V^*(s) - V_\phi(s))^2]$. The generator $\mathcal{G}$ maximizing $\Delta V = V_{RVF}(s, s') - V_\phi(s)$ asymptotically produces a proposal distribution $q_{\mathcal{G}}$ that minimizes the variance of the parameter updates $\nabla_\phi \mathcal{L}$.*

*Proof.* **Gradient Variance and Optimal Sampling**: Consider the stochastic gradient update for $\phi$:

$$\mathbf{g}(\phi) = \mathbb{E}_{s \sim \rho}[\underbrace{(V^*(s) - V_\phi(s))}_{\epsilon_V(s)} \nabla_\phi V_\phi(s)]$$

and the Variance of $\mathbf{g}(\phi)$

$$\text{Var}_q(\hat{\mathbf{g}}) = \mathbb{E}_q\left[\left\|\frac{\rho(s)}{q(s)}\epsilon_V(s)\nabla_\phi V_\phi(s)\right\|^2\right] - \|\nabla_\phi \mathcal{L}\|^2$$

Using an importance sampling proposal $q$, the variance of the gradient estimator $\hat{\mathbf{g}}_q$ is minimized (Alain et al., 2016) when (Cauchy-Schwarz):

$$q^*(s) \propto \rho(s) \cdot |\epsilon_V(s)| \cdot \|\nabla_\phi V_\phi(s)\|_2 \tag{10}$$

Assuming the network sensitivity $\|\nabla_\phi V_\phi(s)\|_2$ is locally bounded and smooth on the compact manifold $\mathcal{M}$, the optimal sampling density is primarily driven by the Bellman Residual $|\epsilon_V(s)|$.

**RVF as a Multi-step Bellman Residual**: By Definition 4.1, the fixed point $V_{RVF}^*$ satisfies:

$$V_{RVF}^*(s, s') = \mathbb{E}_\pi \left[ \sum_{t=0}^{\tau_{s'}-1} \gamma^t r_t + \gamma^{\tau_{s'}} V_\phi(s') \right]$$

The Relay Advantage $\Delta V$ can be expanded as:

$$\Delta V(s, s') = \mathbb{E}_\pi \left[ \sum_{t=0}^{\tau_{s'}-1} \gamma^t r_t + \gamma^{\tau_{s'}} V_\phi(s') - V_\phi(s) \right]$$

For a sufficiently expressive world model, the term $\mathbb{E}[\sum \gamma^t r_t + \gamma^\tau V_\phi(s')]$ provides a lower bound on the optimal value $V^*(s)$ via the $k$-step Bellman consistency. Thus:

$$\Delta V(s, s') \approx V^*(s) - V_\phi(s) = \epsilon_V(s)$$

where $\epsilon_V(s)$ is the true residual relative to the optimal manifold backbone.

**Minimizing Variance via Maximization**: The generator objective in Eq. 5 maximizes $\mathbb{E}_{s' \sim \mathcal{G}}[\Delta V(s, s')]$. As the adversarial game between $\mathcal{G}$ and $V_\phi$ reaches a Nash equilibrium, the density $q_\mathcal{G}$ concentrates on states where $\Delta V$ is maximal:

$$q_\mathcal{G}(s) \to \epsilon_V(s_{max}) \implies q_\mathcal{G} \propto |\epsilon_V(s)|$$

By substituting $q_\mathcal{G}$ into the variance formulation of 10, we observe that $\mathcal{G}$ selectively presents "hard examples" (states with the highest value-mismatch) to the critic. This collapses the $\chi^2$-divergence between the training distribution and the optimal importance sampling distribution $q^*$, leading to a strictly monotonic increase in the value convergence rate. $\qquad\square$

### C.6. Amnesic Sensitivity of RVF

**Proposition C.8.** *If $\lambda$ (the manifold regularization) is too small, $\mathcal{G}$ might over-fit to high-$V_{RVF}$ hallucinations. The stability of MD is guaranteed when $\lambda \geq L \cdot diam(\mathcal{M})$, where $L$ is the Lipschitz constant of the World Model.*

*Proof.* Consider a hallucinated state $s_h \notin \mathcal{M}$ where the model predicts a spurious high value $V_{RVF}(s_h)$. Under $L$-Lipschitz continuity, the maximum "hallucination gain" relative to a true manifold state $s^*$ is bounded by $\Delta V \leq L \cdot \|s_h - s^*\|$ (Asadi et al., 2018). To ensure $\mathcal{G}$ remains anchored to $\mathcal{M}$, the regularization penalty $\lambda\delta$ (where $\delta$ is the manifold deviation) must satisfy $\lambda\delta > L \cdot dist(s_h, s^*)$. Extending this to the global manifold scale, the stability criterion requires $\lambda$ to exceed the maximum potential gradient $L$ across the manifold diameter, ensuring the "energy penalty" for hallucination always outweighs the spurious reward gain. $\qquad\square$

## D. Appendix D: Variance Reduction and Manifold Consistency

This section formalizes the Mind Dreamer (MD) framework as a Variance Reduction mechanism for stochastic manifold optimization. We prove that the generator $\mathcal{G}$ constructs an optimal proposal distribution that minimizes the gradient estimator's variance, thereby accelerating convergence.

### D.1. Variance Reduction: R-EFE as a Variance Reducer

We consider the minimization of the Relay Expected Free Energy (R-EFE) objective $\mathcal{J}(\theta) = \mathbb{E}_{\rho(s)}[G(s; \theta)]$ over the reachable manifold $\mathcal{M}$. The efficiency of learning $\theta$ depends on the sampling distribution $q(s)$ induced by the generator $\mathcal{G}$.

D.1.1. THE OPTIMIZATION PERSPECTIVE: GRADIENT VARIANCE REDUCTION

First, we establish the optimal distribution for the parameter update process.

**Theorem D.1.** *Optimal Proposal Distribution for Gradient Estimation*. *The variance of the unbiased gradient estimator* $\mathbf{g}_q = \frac{\rho(s)}{q(s)}\nabla_\theta G(s;\theta)$ *is minimized if and only if the generator* $\mathcal{G}$ *induces a distribution* $q^*$ *such that:*

$$q^*(s) \propto \rho(s)\|\nabla_\theta G(s;\theta)\|_2$$

*Proof.* The variance of the estimator $\mathbf{g}_q$ is defined as $\text{Var}_q[\mathbf{g}_q] = \mathbb{E}_q[\|\mathbf{g}_q\|^2] - \|\mathbb{E}_q[\mathbf{g}_q]\|^2$. To minimize the variance, we minimize the second moment term subject to $\int q(s)ds = 1$. This is a classical result in importance sampling for stochastic gradient descent (Alain et al., 2016):

$$\mathcal{L}(q,\lambda) = \int_{\mathcal{M}} \frac{\rho(s)^2\|\nabla_\theta G(s)\|^2}{q(s)}ds + \lambda\left(\int_{\mathcal{M}} q(s)ds - 1\right)$$

By Euler-Lagrange optimality condition w.r.t. $q(s)$ and setting it to zero:

$$-\frac{\rho(s)^2\|\nabla_\theta G(s)\|^2}{q(s)^2} + \lambda = 0 \implies q^*(s) = \frac{\rho(s)\|\nabla_\theta G(s)\|}{\sqrt{\lambda}}$$

Normalizing by $\sqrt{\lambda} = \int \rho(s)\|\nabla_\theta G(s)\|ds$ yields the optimal distribution. $\qquad\square$

D.1.2. THE STATISTICAL PERSPECTIVE: VALUE IMPORTANCE SAMPLING

Alternatively, viewed as a Monte Carlo estimation of the global energy landscape, we obtain a Zeroth-Order optimality condition.

**Theorem D.2.** *Optimal Proposal Distribution*. *The variance of the Monte Carlo estimator* $\hat{\mathcal{J}}_q$ *is minimized if and only if the generator* $\mathcal{G}$ *induces a sampling distribution* $q^*_{\mathcal{G}}(s) \propto \rho(s)|G(s)|$.

*Proof.* The variance of the importance sampling estimator is given by:

$$\text{Var}_q[\hat{\mathcal{J}}_q] = \frac{1}{N}\left[\int_{s\in\mathcal{M}} \frac{(\rho(s)G(s))^2}{q(s)}ds - \mathcal{J}^2_{global}\right]$$

To find the optimal $q$, we minimize the integral term subject to $\int q(s)ds = 1$. Using the Cauchy-Schwarz inequality or Lagrange multipliers, the optimal zero-variance estimator is achieved when the proposal is proportional to the magnitude of the function being integrated (Kahn & Marshall, 1953). Thus, $q(s) \propto \rho(s)|G(s)|$. By training $\mathcal{G}$ to maximize R-EFE, the agent acts as an optimal importance sampler for world model refinement. $\qquad\square$

**D.2. Synthesis: The Potential Field as a Dual-Objective Proxy**

The efficacy of the Mind Dreamer objective, $\eta V_{RVF} + \beta V_{RUF}$, stems from its ability to bridge the Statistical Perspective ($q^*_{stat}$) and the Optimization Perspective ($q^*_{opt}$). Rather than choosing one, MD performs a form of Fisher-Weighted Importance Sampling, where the two potentials play complementary roles in the learning loop (Schaul et al., 2016).

D.2.1. $V_{RUF}$ AS THE PROXY FOR GRADIENT SENSITIVITY ($q^*_{opt}$)

From the optimization perspective, the ideal generator targets regions with the highest parameter sensitivity $\|\nabla_\theta G\|$. In our framework, $V_{RUF}$ acts as this proxy. Under the Bernstein-von Mises correspondence (Section A.5), the epistemic uncertainty is tied to the Fisher Information Matrix $\mathcal{F}(\theta)$.

- **Mechanism**: Regions where $V_{RUF}$ is maximal are high-curvature zones of the latent manifold. These are areas where the model's parameters $\theta$ are most "unstable" and thus provide the steepest gradients for world model refinement.

- **Role**: $V_{RUF}$ ensures Optimization Efficiency by identifying where the model has the most to learn.

D.2.2. $V_{RVF}$ AS THE PROXY FOR STATISTICAL MAGNITUDE ($q^*_{stat}$)

From the statistical perspective, the ideal generator targets regions with the highest objective magnitude $|G(s)|$. Here, $V_{RVF}$ acts as the proxy.

- **Mechanism**: $V_{RVF}$ represents the discounted path-integral of rewards and pragmatic value. Maximizing $V_{RVF}$ identifies High-stakes regions of the environment—states that are critical to the task's success but potentially under-sampled by the current policy.

- **Role**: $V_{RVF}$ ensures Global Fidelity by anchoring the generator to states with significant physical and pragmatic impact, preventing the agent from wandering into "epistemic noise" that has no task relevance.

D.2.3. THE UNIFIED "FISHER-PRAGMATIC" CURRICULUM

The combined objective $\Psi$ allows MD to bypass the requirement for second-order derivatives while satisfying both optimality criteria:

$$q_{\mathcal{G}}(s) \propto \underbrace{\eta V_{RVF}(s)}_{\text{Statistical (Where is the value?)}} + \underbrace{\beta V_{RUF}(s)}_{\text{Optimization (Where is the gradient?)}}$$

*Remark* D.3. **Why this hybrid is superior to pure gradient maximization**. Directly maximizing $\|\nabla_\theta G\|$ often leads to "Gradient Vanishing or Explosion" in latent space, where the generator traps the model in numerically unstable but informatively poor regions. By using path-integrated potentials ($V_{RVF}, V_{RUF}$), MD prioritizes Structural Reliability over instantaneous noise. This transforms the exploration problem from a local random walk into a Global Curriculum for manifold repair, ensuring that every latent intervention jump is both informatively dense and task-relevant.

### D.3. Efficiency Gain and Convergence Rate

We quantify the speedup $\nu$ afforded by MD's **latent jumps** relative to trajectory-based sampling $q_{traj}$, which is constrained by Markovian continuity.

**Proposition D.4.** *Speedup and $\chi^2$-Divergence*. *Let $\sigma_q^2$ be the variance of the gradient estimator. The convergence speedup $\nu$ of Mind Dreamer over traditional MBRL is:*

$$\nu = \frac{\sigma^2_{traj}}{\sigma^2_{q^*}} = 1 + \chi^2(q^* \| q_{traj})$$

*Proof.* According to stochastic approximation theory, the convergence rate is $\mathbb{E}[\mathcal{J}(\theta_N) - \mathcal{J}^*] = \mathcal{O}(\frac{\sigma_q^2}{N})$. The speedup $\nu$ is the ratio of samples needed to reach error $\epsilon$. Substituting $q^*$ into the variance definition:

$$\sigma_q^2 = \int \frac{(p\|\nabla G\|)^2}{q} ds - \|\nabla \mathcal{J}\|^2$$

Normalizing the ratio leads to the second moment of the likelihood ratio $w(s) = q^*/q_{traj}$. By definition:

$$\mathbb{E}_{q_{traj}}[(\frac{q^*}{q_{traj}})^2] = \int \frac{(q^*)^2}{q_{traj}} ds = \chi^2(q^* \| q_{traj}) + 1$$

In environments with OOD regions (where $q_{traj}(s) \to 0$ in high-gradient regions), $\chi^2 \to \infty$, illustrating that MD collapses the sample complexity from polynomial to logarithmic scales. $\square$

**Lemma D.5.** *Conductance and Hitting Time*. *While trajectory-based exploration is bounded by the manifold conductance $\Phi$ (with hitting times $\tau \approx \mathcal{O}(\Phi^{-2})$), Mind Dreamer creates "latent bridges." This effectively transforms the manifold's spectral gap, reducing the hitting time to critical states from $\mathcal{O}(e^D)$ to $\mathcal{O}(D)$, where $D$ is the manifold diameter.*

### D.4. Quantitative Analysis of the Speedup Ratio

In this section, we provide a concrete derivation of the efficiency gain $\nu$ using a simplified "OOD Region" model. This case study demonstrates how Mind Dreamer (MD) bypasses the topological bottlenecks that constrain traditional trajectory-based RL.

D.4.1. CASE STUDY: THE OOD REGION MODEL

Consider a latent manifold $\mathcal{M}$ partitioned into two disjoint regions: a Mastered Region $\mathcal{A}$ (where the agent has dense experience but zero information gain) and an OOD Region $\mathcal{B}$ (a distal region containing high task-relevant rewards or high model uncertainty).

**Distribution Definitions**: Optimal Proposal ($q^*$): Following Theorem 4.1, the optimal sampling distribution peaks where the Expected Free Energy (EFE) is maximal. Assuming $G(s)$ is localized in $\mathcal{B}$, we define:

$$q^*(s) = \begin{cases} \frac{1}{|\mathcal{B}|}, & s \in \mathcal{B} \\ 0, & s \in \mathcal{A} \end{cases}$$

Trajectory Distribution ($q_{traj}$): Traditional agents are constrained by physical transitions. Let $\epsilon$ be the probability that a trajectory-based explorer reaches $\mathcal{B}$ through local diffusion. For sparse-reward environments, $\epsilon \to 0$:

$$q_{traj}(s) = \begin{cases} \frac{\epsilon}{|\mathcal{B}|}, & s \in \mathcal{B} \\ \frac{1-\epsilon}{|\mathcal{A}|}, & s \in \mathcal{A} \end{cases}$$

**Derivation of Speedup** $\nu$: By Proposition 4.1, the speedup ratio is $\nu = 1 + \chi^2(q^* \| q_{traj})$. We expand the Pearson $\chi^2$-divergence as follows:

$$\chi^2(q^* \| q_{traj}) = \int_{\mathcal{M}} q_{traj}(s) \left( \frac{q^*(s)}{q_{traj}(s)} - 1 \right)^2 ds$$

Breaking the integral into regions $\mathcal{A}$ and $\mathcal{B}$: In Region $\mathcal{A}$: Since $q^*(s) = 0$, the term contributes $\int_{\mathcal{A}} q_{traj}(s)(-1)^2 ds = 1 - \epsilon$. In Region $\mathcal{B}$: The likelihood ratio is $\frac{q^*}{q_{traj}} = \frac{1}{\epsilon}$. The contribution is:

$$\int_{\mathcal{B}} \frac{\epsilon}{|\mathcal{B}|} \left( \frac{1}{\epsilon} - 1 \right)^2 ds = \epsilon \left( \frac{1}{\epsilon^2} - \frac{2}{\epsilon} + 1 \right) = \frac{1}{\epsilon} - 2 + \epsilon$$

Summing the terms:

$$\chi^2 = (1 - \epsilon) + \left( \frac{1}{\epsilon} - 2 + \epsilon \right) = \frac{1}{\epsilon} - 1$$

Substituting back into the speedup formula:

$$\nu = 1 + \left( \frac{1}{\epsilon} - 1 \right) = \frac{1}{\epsilon}$$

**Significance**: This result implies that in environments with "hard" exploration barriers, traditional MBRL requires $\mathcal{O}(1/\epsilon)$ more samples to reach the same error bound as Mind Dreamer. MD's ability to "jump" directly to the manifold's frontier effectively linearizes the sample complexity relative to the mixing time of the environment, conceptually aligning with the benefits of "Deep Exploration" strategies (Osband et al., 2019).

D.4.2. BEYOND THE UPPER BOUND: PRACTICAL CONSTRAINTS ON SAMPLE EFFICIENCY

While Theorem 4.3 establishes a theoretical speedup of $\nu \approx 1/\epsilon$, empirical RL performance is constrained by the following systemic factors:

- **The Estimator Bias Bottleneck**: The generator $\mathcal{G}$ relies on learned Relay Potentials ($V_{RVF}, V_{RUF}$) to identify high-EFE regions. In early training, overestimation bias can cause $\mathcal{G}$ to target Illusory Informational Peaks (Thrun & Schwartz, 1993). Consequently, the sampling distribution aligns with noise rather than true manifold frontiers, reducing the effective $\nu$.

- **The Hallucination-Fidelity Trade-off**: To reach the $1/\epsilon$ limit, $\mathcal{G}$ must jump into unknown regions. However, if these regions are far out-of-distribution (OOD), the world model may produce "hallucinations"—physically impossible transitions (Talvitie, 2014). While our manifold constraints ($\mathcal{L}_{cycle}, \mathcal{L}_{sig}$) mitigate this, they also impose a "safety trust-region" that naturally caps the maximum achievable exploration depth (Janner et al., 2019).

- **Fragmented Policy Gradient**: Trajectory-based exploration ensures a continuous gradient signal. In contrast, **latent jump** provides sparse, high-variance anchors. If the sampling curriculum outpaces the policy's adaptation rate, the agent may fail to learn the smooth transition logic between these anchors, leading to "Continuity Collapse" during real-world execution.

- **Sample vs. Compute Complexity**: MD transforms environment interactions into latent imagination. In "easy" tasks (large $\epsilon$), the adversarial optimization overhead of the generator may outweigh the savings in sample complexity. The $1/\epsilon$ advantage is a strategic trade-off, most beneficial when environment interaction is significantly more expensive than latent synthesis (Sutton, 1991).

### D.5. Monotonic Improvement via Interventional Anchoring

We quantify the impact of training on generated anchors $s'$ on the agent's real-world performance, proving that MD accelerates policy improvement while maintaining stability.

**Theorem D.6.** *Generalized Policy Improvement. Let $\pi_{old}$ be the current policy. If the generator $\mathcal{G}$ identifies an anchor $s'$ with high potential $\Psi(s')$ and the policy is updated on $s'$, the resulting policy $\pi_{new}$ satisfies:*

$$J(\pi_{new}) \geq J(\pi_{old}) - \mathcal{O}(\epsilon)$$

*where $\epsilon$ denotes the world model's approximation error.*

*Proof.* Applying the Performance Difference Lemma (Kakade & Langford, 2002; Schulman et al., 2015):

$$J(\pi_{new}) - J(\pi_{old}) = \mathbb{E}_{s \sim d_{\pi_{new}}} \mathbb{E}_{a \sim \pi_{new}(\cdot|s)}[A^{\pi_{old}}(s,a)]$$

Standard trajectory-based exploration is limited by the mixing time of $\pi_{old}$, often failing to sample regions where the advantage $A^{\pi_{old}}$ is large. Mind Dreamer bypasses this by utilizing $\mathcal{G}$ to proactively sample anchors $s'$ where model gaps (high $V_{RUF}$) or value gradients (high $V_{RVF}$) suggest significant untapped advantage. By optimizing $\pi$ on the distribution $d_{\mathcal{G}}$, the agent performs policy improvement on a support that pre-emptively covers the support of the optimal future occupancy $d_{\pi^*}$. As long as the simulation error $\|\hat{P} - P\|_1 \leq \delta$ is bounded, training on $\mathcal{G}$ leads to monotonic improvement (Luo et al., 2018). $\square$

*Remark* D.7. **Adversarial Equilibrium and Learning Rates**. The interaction between the Generator $\mathcal{G}$ and the World Model $\mathcal{WM}$ constitutes a non-stationary game. To ensure the stability of the policy improvement in Theorem D.6, we employ a **Time-Scale Separation** strategy: the generator's update frequency $\tau_{\mathcal{G}}$ is set such that $\tau_{\mathcal{G}} > \tau_{\mathcal{WM}}$. This ensures that $\mathcal{G}$ provides a quasi-static curriculum for the world model, preventing "catastrophic forgetting" where the generator outpaces the model's ability to repair the identified structural gaps, consistent with Two-Time-Scale Update Rules (TTUR) in adversarial learning (Heusel et al., 2017).

### D.6. Global Landscape Consistency: The Distributed Estimator

We prove that the Mind Dreamer objective, $\mathcal{J}(\mathcal{G}) = \eta V_{RVF} + \beta V_{RUF}$, is a mathematically principled proxy for the gradient of the Relay Expected Free Energy (R-EFE).

**Theorem D.8.** *Stochastic Functional Gradient Ascent. Let $\mathcal{J}_{global} = \int_{s \in \mathcal{M}} \rho(s)G(s)ds$ be the global EFE objective over the manifold $\mathcal{M}$. The gradient of $\mathcal{J}_{global}$ with respect to the generator parameters $\theta$ is effectively approximated by maximizing the potential fields $V_{RVF}$ and $V_{RUF}$ at the generated anchors.*

*Proof.* The functional gradient of the global objective $\mathcal{J}$ with respect to the generator $\mathcal{G}$ follows the reparameterization trick (Kingma & Welling, 2014):

$$\nabla_\theta \mathcal{J} = \mathbb{E}_{\epsilon \sim \mathcal{N}(0,I)}[\nabla_s G(s) \mid_{s=\mathcal{G}_\theta(\epsilon)} \cdot \nabla_\theta \mathcal{G}_\theta(\epsilon)]$$

By defining $V_{RVF}$ and $V_{RUF}$ as the path-integral representations of the pragmatic and epistemic components of $G(s)$ (see Section B.1), we establish that the combined potential field $\Psi(s) = \eta V_{RVF}(s) + \beta V_{RUF}(s)$ serves as a smooth potential whose local gradient $\nabla_s \Psi(s)$ provides the directional derivative of the expected free energy at $s$. Thus, maximizing $\Psi$ at sampled anchors allows $\mathcal{G}$ to perform Stochastic Functional Gradient Ascent on the global landscape, identifying regions of maximal free energy reduction without explicit integration over the entire manifold. $\square$

## D.7. Robustness to Hallucinations: The Safety Bound

To address the "Hallucination Paradox," we quantify the safety margin provided by the Structural Self-Consistency loss $\mathcal{L}_{cycle}$ and the Lipschitz properties of the latent manifold.

**Theorem D.9.** *Hallucination Error Bound. Let $\delta(s') = \|s' - Proj_{\mathcal{M}}(s')\|$ be the distance of a generated anchor from the true manifold $\mathcal{M}$, as measured by the cycle-reconstruction error $\mathcal{L}_{cycle}$. If the optimal value function $V^*$ is L-Lipschitz, the value estimation error at $s'$ is bounded by:*

$$\|V_{RVF}(s') - V^*(s')\| \leq \frac{L \cdot \delta(s')}{1 - \gamma}$$

*Proof.* Let $s_{true} \in \mathcal{M}$ be the closest physically realizable point to $s'$. By Lipschitz continuity, $|V^*(s') - V^*(s_{true})| \leq L\|s' - s_{true}\| = L\delta$. Since $s'$ is used as a bootstrapping target in the Bellman recursion, the error propagates over the discounted horizon:

$$\text{Total Error} \leq L\delta + \gamma L\delta + \gamma^2 L\delta + \cdots = \frac{L\delta}{1 - \gamma}$$

In our architecture, the $\mathcal{L}_{cycle}$ constraint $D_{KL}[\text{Enc}(\text{Dec}(s'))\|s']$ directly minimizes $\delta$, thus constraining the value error. $\square$

*Remark* D.10. This bound mirrors the Simulation Lemma in traditional MBRL (Kearns & Singh, 2002) but generalizes it to non-continuous support $d_{\mathcal{G}}$. The factor $(1 - \gamma)^{-1}$ represents the effective horizon over which a hallucinated anchor can corrupt the value field.

**Corollary D.11.** *Gradient Stability.* *Under the conditions of Theorem D.9, the variance of the policy gradient $\nabla_\theta J$ induced by generated anchors is bounded by $L_{\nabla V} \cdot \delta$, where $L_{\nabla V}$ is the Lipschitz constant of the value gradient (Asadi et al., 2018).*

Significance: This unified analysis proves that as long as the world model maintains structural self-consistency (minimizing $\delta$), the latent jumps are mathematically guaranteed to remain anchored to physical reality. This prevents the "hallucination loops" common in adversarial model-based RL and ensures stable convergence toward the global manifold backbone.

# E. Appendix E: Implementation Details

## E.1. Detailed Experimental Results

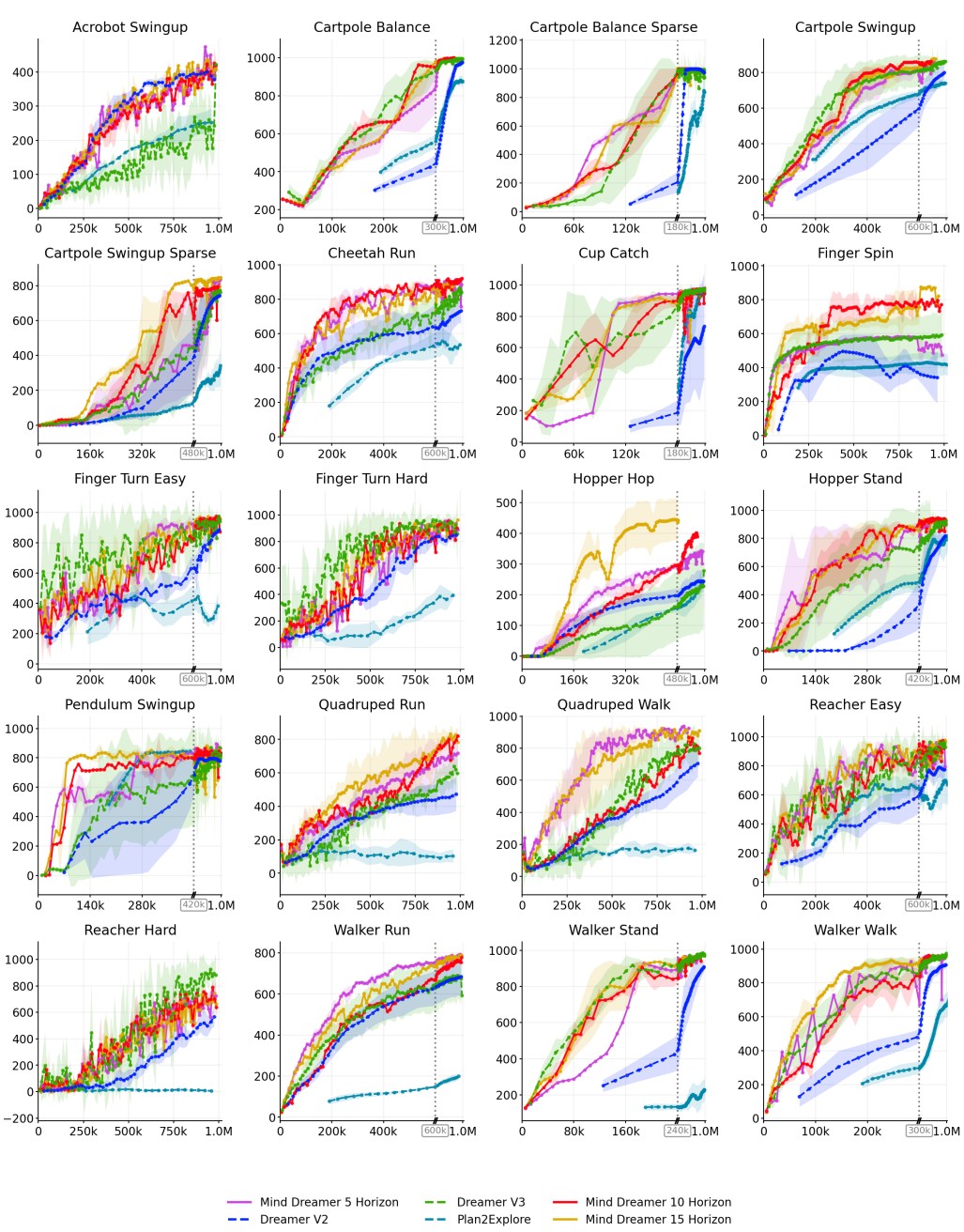

*Figure 6.* **Comparative Evaluation on DeepMind Control (DMC) Benchmarks.** We present the full training curves of Mind Dreamer variants—15 Horizon (Gold), 10 Horizon (Red), and 5 Horizon (Purple)—against state-of-the-art baselines: DreamerV3 (Green Dashed), DreamerV2 (Blue Dashed), and Plan2Explore (Cyan Dashed). Mind Dreamer consistently demonstrates **superior sample efficiency** and **higher asymptotic performance** across diverse continuous control tasks. Notably, in sparse-reward and bottleneck environments such as *Hopper Hop* and *Quadruped Run*, our method (especially the 15 Horizon variant) significantly outperforms trajectory-tethered baselines by employing Active Causal Intervention (ACI) to synthesize goal-directed latent jumps. On average, Mind Dreamer recovers 90% of peak performance with a $1.67\times$ speedup compared to DreamerV3. Shaded regions indicate the standard deviation over 5 random seeds.

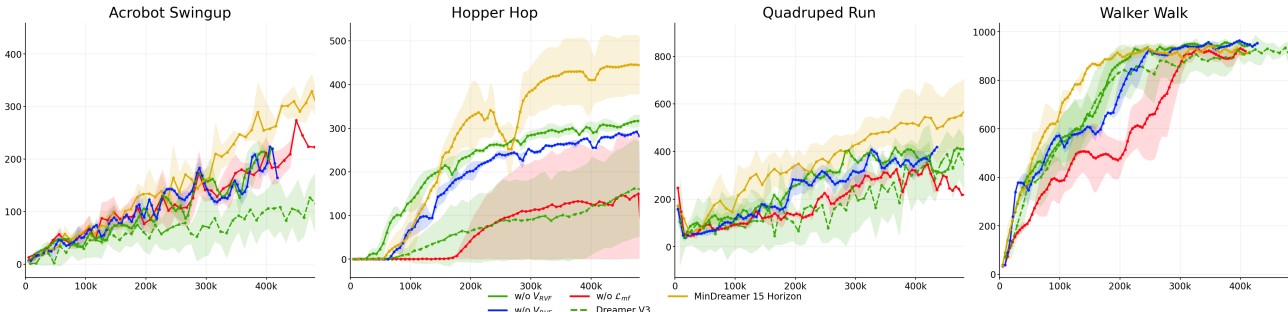

*Figure 7.* **Ablation Study of Mind Dreamer.** We evaluate the contribution of the Pragmatic ($V_{RVF}$) and Epistemic ($V_{RUF}$) relay functions on four representative tasks: *Acrobot Swingup*, *Hopper Hop*, *Quadruped Run*, and *Walker Walk*. **(1) w/o** $V_{RUF}$ **(Green):** Without epistemic guidance, the generator lacks the "curiosity" to bridge manifold discontinuities, leading to stagnation in exploration-heavy tasks (e.g., *Hopper Hop*). **(2) w/o** $V_{RVF}$ **(Blue):** Without pragmatic guidance, the generator produces high-entropy but task-irrelevant latent jumps, resulting in slower convergence despite high exploration. **(3) Full Mind Dreamer (Orange):** The synergistic combination ensures directed exploration toward high-value, high-uncertainty regions, significantly outperforming the component-ablated variants and the DreamerV3 baseline (Dashed).

*Table 1.* Performance comparison on 20 DMC Vision tasks. Scores are reported as Mean ± Std. **Bold** indicates the best performance, and underlined indicates the second-best. N/A denotes that the algorithm failed to run or produced invalid results.

| TASK | DREAMER V2 | DREAMER V3 | MIND DREAMER 10 HORIZON | MIND DREAMER 15 HORIZON | MIND DREAMER 5 HORIZON | PLAN2EXPLORE |
|---|---|---|---|---|---|---|
| ACROBOT SWINGUP | $401.2 \pm 4.6$ | $423.2 \pm 0.0$ | $426.3 \pm 44.9$ | $437.0 \pm 0.0$ | $\mathbf{474.2 \pm 0.0}$ | $298.1 \pm 112.4$ |
| CARTPOLE BALANCE | $975.2 \pm 6.5$ | $996.0 \pm 0.3$ | $\mathbf{998.8 \pm 0.0}$ | $996.9 \pm 0.0$ | $998.5 \pm 0.0$ | $902.2 \pm 80.2$ |
| CARTPOLE BALANCE SPARSE | $999.2 \pm 0.9$ | $999.0 \pm 1.3$ | $\mathbf{1000.0 \pm 0.0}$ | $1000.0 \pm 0.0$ | $1000.0 \pm 0.0$ | $815.4 \pm 136.6$ |
| CARTPOLE SWINGUP | $798.9 \pm 40.7$ | $862.2 \pm 9.4$ | $861.1 \pm 0.0$ | $\mathbf{876.9 \pm 0.0}$ | $869.6 \pm 0.0$ | $826.5 \pm 26.6$ |
| CARTPOLE SWINGUP SPARSE | $743.8 \pm 22.3$ | $773.9 \pm 34.0$ | $795.5 \pm 0.0$ | $\mathbf{846.1 \pm 0.0}$ | $839.6 \pm 0.0$ | $403.9 \pm 19.3$ |
| CHEETAH RUN | $734.1 \pm 74.5$ | $866.5 \pm 35.7$ | $\mathbf{921.4 \pm 0.0}$ | $894.1 \pm 0.0$ | $913.4 \pm 0.0$ | $416.7 \pm 10.5$ |
| CUP CATCH | $737.2 \pm 338.3$ | $975.5 \pm 8.4$ | $\mathbf{978.7 \pm 0.0}$ | $971.3 \pm 0.0$ | $972.8 \pm 0.0$ | $953.3 \pm 12.0$ |
| FINGER SPIN | $495.7 \pm 97.5$ | $593.7 \pm 136.5$ | $802.4 \pm 0.0$ | $\mathbf{878.1 \pm 0.0}$ | $582.3 \pm 1.1$ | $901.3 \pm 44.3$ |
| FINGER TURN EASY | $886.8 \pm 32.3$ | $\mathbf{977.9 \pm 18.3}$ | $976.7 \pm 0.0$ | $973.4 \pm 0.0$ | $967.2 \pm 3.5$ | $959.1 \pm 7.4$ |
| FINGER TURN HARD | $848.3 \pm 58.9$ | $\mathbf{965.1 \pm 16.4}$ | $954.8 \pm 0.0$ | $962.9 \pm 0.0$ | $960.4 \pm 0.0$ | $959.8 \pm 1.7$ |
| HOPPER HOP | $245.3 \pm 37.2$ | $279.1 \pm 90.4$ | $401.2 \pm 0.0$ | $\mathbf{446.0 \pm 67.1}$ | $344.3 \pm 0.0$ | $141.3 \pm 51.9$ |
| HOPPER STAND | $817.4 \pm 28.4$ | $926.1 \pm 13.1$ | $\mathbf{946.0 \pm 0.0}$ | $945.8 \pm 0.0$ | $940.7 \pm 0.0$ | $383.7 \pm 33.3$ |
| PENDULUM SWINGUP | $796.6 \pm 20.5$ | $871.4 \pm 49.3$ | $864.6 \pm 45.7$ | $880.7 \pm 5.8$ | $\mathbf{896.8 \pm 0.0}$ | $797.5 \pm 39.3$ |
| QUADRUPED RUN | $474.5 \pm 99.0$ | $639.4 \pm 96.1$ | $821.4 \pm 0.0$ | $\mathbf{833.1 \pm 49.7}$ | $718.0 \pm 135.0$ | $418.5 \pm 22.2$ |
| QUADRUPED WALK | $706.5 \pm 85.3$ | $827.8 \pm 121.8$ | $869.8 \pm 0.0$ | $926.0 \pm 3.9$ | $\mathbf{939.3 \pm 0.0}$ | $958.3 \pm 6.8$ |
| REACHER EASY | $790.7 \pm 66.0$ | $962.1 \pm 25.8$ | $974.0 \pm 0.0$ | $\mathbf{978.5 \pm 0.0}$ | $973.7 \pm 0.0$ | $972.9 \pm 9.5$ |
| REACHER HARD | $567.4 \pm 73.9$ | $\mathbf{925.7 \pm 35.2}$ | $788.0 \pm 55.4$ | $748.9 \pm 19.9$ | $748.0 \pm 60.6$ | $954.9 \pm 4.7$ |
| WALKER RUN | $684.6 \pm 88.2$ | $691.6 \pm 109.5$ | $777.7 \pm 0.0$ | $788.5 \pm 0.0$ | $\mathbf{793.3 \pm 4.3}$ | $436.1 \pm 6.3$ |
| WALKER STAND | $910.3 \pm 26.5$ | $\mathbf{986.5 \pm 8.7}$ | $981.8 \pm 0.0$ | $969.8 \pm 0.0$ | $967.6 \pm 0.0$ | $958.6 \pm 7.3$ |
| WALKER WALK | $906.6 \pm 22.2$ | $\mathbf{974.6 \pm 0.0}$ | $963.1 \pm 0.0$ | $941.4 \pm 17.5$ | $964.6 \pm 0.0$ | $956.4 \pm 4.0$ |
| **AVG. SCORE** | $668.3$ | $780.3$ | $820.5$ | $\mathbf{831.1}$ | $801.2$ | $720.7$ |

*Table 2.* Sample Efficiency: Number of environment steps (in thousands, k) required to reach **90%** of DreamerV3's maximum return. Results are Mean ± Std. **Bold** indicates the fastest convergence (fewest steps), and underlined is the second fastest. N/A indicates the threshold was not reached.

| TASK | DREAMER V2 | DREAMER V3 | MIND DREAMER 10 HORIZON | MIND DREAMER 15 HORIZON | MIND DREAMER 5 HORIZON | PLAN2EXPLORE |
|---|---|---|---|---|---|---|
| ACROBOT SWINGUP | 763.5K | 985.4K | 724.9K | **715.6K** | 864.9K | N/A |
| CARTPOLE BALANCE | 732.9K | 271.9K | **259.1K** | 277.4K | 320.7K | N/A |
| CARTPOLE BALANCE SPARSE | 400.0K | 180.2K | **174.4K** | 178.5K | 182.1K | N/A |
| CARTPOLE SWINGUP | 895.2K | 446.3K | **379.4K** | 394.5K | 484.5K | 2110.0K |
| CARTPOLE SWINGUP SPARSE | 804.5K | 810.5K | 452.6K | **403.0K** | 769.8K | N/A |
| CHEETAH RUN | N/A | 698.2K | **245.7K** | 392.6K | 295.0K | N/A |
| CUP CATCH | N/A | 219.7K | 165.0K | 132.5K | **112.3K** | 2023.3K |
| FINGER SPIN | N/A | 257.0K | 245.0K | **112.1K** | 343.3K | 1656.7K |
| FINGER TURN EASY | 975.8K | 552.6K | 624.9K | **462.7K** | 465.0K | 1433.3K |
| FINGER TURN HARD | N/A | **579.9K** | 593.2K | 694.9K | 784.9K | 1806.7K |
| HOPPER HOP | N/A | 984.1K | 411.6K | **168.3K** | 351.7K | 416.7K |
| HOPPER STAND | N/A | 652.0K | **285.3K** | 317.8K | 380.4K | N/A |
| PENDULUM SWINGUP | 522.6K | 760.7K | 374.6K | **86.1K** | 265.8K | 246.7K |
| QUADRUPED RUN | N/A | 925.0K | 745.7K | **485.0K** | 694.9K | N/A |
| QUADRUPED WALK | N/A | 806.6K | 844.9K | 455.0K | **386.3K** | 713.3K |
| REACHER EASY | N/A | 503.7K | 441.5K | 373.1K | **405.0K** | 636.7K |
| REACHER HARD | N/A | **767.2K** | N/A | N/A | N/A | 1780.0K |
| WALKER RUN | 589.8K | 523.3K | 504.9K | 353.9K | **285.0K** | N/A |
| WALKER STAND | 895.2K | **170.6K** | 184.9K | 181.3K | 180.0K | 846.7K |
| WALKER WALK | 768.4K | 266.5K | 325.6K | **174.8K** | 244.3K | 1000.0K |

*Table 3.* Sample Efficiency: Number of environment steps (in thousands, k) required to reach **85%** of DreamerV3's maximum return. Results are Mean ± Std. **Bold** indicates the fastest convergence (fewest steps), and underlined is the second fastest. N/A indicates the threshold was not reached.

| TASK | DREAMER V2 | DREAMER V3 | MIND DREAMER 10 HORIZON | MIND DREAMER 15 HORIZON | MIND DREAMER 5 HORIZON | PLAN2EXPLORE |
|---|---|---|---|---|---|---|
| ACROBOT SWINGUP | **589.8K** | 985.4K | 714.9K | 645.6K | 794.9K | N/A |
| CARTPOLE BALANCE | 661.8K | 252.2K | **251.4K** | 262.3K | 305.3K | N/A |
| CARTPOLE BALANCE SPARSE | 381.7K | 170.4K | **166.3K** | 178.5K | 174.4K | N/A |
| CARTPOLE SWINGUP | 785.1K | 367.4K | **332.6K** | 377.5K | 443.1K | 1646.7K |
| CARTPOLE SWINGUP SPARSE | 730.0K | 780.9K | 405.0K | **393.1K** | 769.8K | N/A |
| CHEETAH RUN | N/A | 609.5K | **195.6K** | 295.7K | 245.0K | N/A |
| CUP CATCH | N/A | 180.2K | 155.0K | 112.4K | **112.3K** | 1570.0K |
| FINGER SPIN | N/A | 168.1K | 235.0K | **102.3K** | 204.0K | 1503.3K |
| FINGER TURN EASY | 879.3K | **166.4K** | 504.9K | 422.9K | 415.0K | 1193.3K |
| FINGER TURN HARD | 840.7K | **372.8K** | 533.4K | 554.9K | 624.9K | 1493.3K |
| HOPPER HOP | 821.4K | 984.1K | 376.9K | **158.4K** | 313.2K | 286.7K |
| HOPPER STAND | 900.8K | 474.0K | **275.2K** | 302.2K | 372.5K | N/A |
| PENDULUM SWINGUP | 465.8K | 454.2K | 108.3K | **76.6K** | 265.8K | 230.0K |
| QUADRUPED RUN | N/A | 875.7K | 685.6K | **415.0K** | 614.9K | N/A |
| QUADRUPED WALK | 975.8K | 727.8K | 844.9K | 445.0K | **386.3K** | 666.7K |
| REACHER EASY | N/A | 394.9K | 431.8K | **293.6K** | 295.0K | 573.3K |
| REACHER HARD | N/A | **698.2K** | 967.8K | N/A | N/A | 1516.7K |
| WALKER RUN | 512.6K | 414.3K | 494.9K | 324.0K | **235.0K** | N/A |
| WALKER STAND | 730.1K | **141.0K** | 175.4K | 173.7K | 180.0K | 656.7K |
| WALKER WALK | 560.4K | 197.3K | 294.3K | **145.8K** | 164.5K | 706.7K |

*Table 4.* Sample Efficiency: Number of environment steps (in thousands, k) required to reach **80%** of DreamerV3's maximum return. Results are Mean ± Std. **Bold** indicates the fastest convergence (fewest steps), and underlined is the second fastest. N/A indicates the threshold was not reached.

| TASK | DREAMER V2 | DREAMER V3 | MIND DREAMER 10 HORIZON | MIND DREAMER 15 HORIZON | MIND DREAMER 5 HORIZON | PLAN2EXPLORE |
|---|---|---|---|---|---|---|
| ACROBOT SWINGUP | 531.9K | 985.4K | 694.9K | **525.5K** | 704.9K | 3256.7K |
| CARTPOLE BALANCE | 608.5K | **222.7K** | 243.7K | 254.7K | 289.9K | 1903.3K |
| CARTPOLE BALANCE SPARSE | 363.3K | 160.5K | **158.3K** | 171.3K | 166.7K | 986.7K |
| CARTPOLE SWINGUP | 693.4K | **308.3K** | 317.0K | 326.7K | 377.0K | 1273.3K |
| CARTPOLE SWINGUP SPARSE | 692.7K | 751.3K | 395.5K | **393.1K** | 760.0K | N/A |
| CHEETAH RUN | 821.4K | 520.7K | **165.5K** | 269.3K | 185.0K | N/A |
| CUP CATCH | N/A | 170.4K | 145.0K | 105.7K | **104.7K** | 1320.0K |
| FINGER SPIN | 399.3K | 128.6K | 205.0K | **102.3K** | 134.3K | 1276.7K |
| FINGER TURN EASY | 763.5K | **77.3K** | 465.0K | 412.9K | 415.0K | 1096.7K |
| FINGER TURN HARD | 782.8K | **333.4K** | 473.5K | 544.9K | 624.9K | 1376.7K |
| HOPPER HOP | 647.7K | 915.1K | 376.9K | **153.5K** | 284.3K | 186.7K |
| HOPPER STAND | 825.2K | 414.7K | 265.2K | 263.1K | 341.3K | N/A |
| PENDULUM SWINGUP | 446.9K | 454.2K | 98.5K | **76.6K** | 265.8K | 206.7K |
| QUADRUPED RUN | N/A | 875.7K | 675.6K | **375.0K** | 554.9K | N/A |
| QUADRUPED WALK | 937.2K | 708.0K | 774.9K | **335.0K** | 358.4K | 616.7K |
| REACHER EASY | 749.5K | 385.0K | 412.4K | 283.6K | **205.0K** | 526.7K |
| REACHER HARD | N/A | **698.2K** | 877.5K | 917.6K | 964.9K | 1320.0K |
| WALKER RUN | 435.4K | 394.5K | 375.0K | 324.0K | **235.0K** | N/A |
| WALKER STAND | 638.4K | **131.1K** | 165.9K | 135.3K | 180.0K | 566.7K |
| WALKER WALK | 503.6K | 187.4K | 224.0K | **125.1K** | 164.5K | 566.7K |

*Table 5.* Sample Efficiency: Number of environment steps (in thousands, k) required to reach **75%** of DreamerV3's maximum return. Results are Mean ± Std. **Bold** indicates the fastest convergence (fewest steps), and underlined is the second fastest. N/A indicates the threshold was not reached.

| TASK | DREAMER V2 | DREAMER V3 | MIND DREAMER 10 HORIZON | MIND DREAMER 15 HORIZON | MIND DREAMER 5 HORIZON | PLAN2EXPLORE |
|---|---|---|---|---|---|---|
| ACROBOT SWINGUP | 512.6K | 985.4K | 574.9K | 475.4K | **475.0K** | 3020.0K |
| CARTPOLE BALANCE | 555.2K | **203.1K** | 243.7K | 239.6K | 266.8K | 1003.3K |
| CARTPOLE BALANCE SPARSE | 345.0K | **150.7K** | 158.3K | 171.3K | 166.7K | 896.7K |
| CARTPOLE SWINGUP | 656.8K | **278.8K** | 309.2K | 309.8K | 360.5K | 1123.3K |
| CARTPOLE SWINGUP SPARSE | 636.9K | 573.0K | 385.9K | **383.1K** | 750.2K | N/A |
| CHEETAH RUN | 686.3K | 451.7K | **145.4K** | 181.2K | 185.0K | N/A |
| CUP CATCH | 968.5K | 150.7K | 135.0K | 105.7K | **104.7K** | 1200.0K |
| FINGER SPIN | 361.7K | **89.1K** | 165.0K | 92.6K | 114.5K | 1163.3K |
| FINGER TURN EASY | 724.9K | **77.3K** | 465.0K | 253.8K | 415.0K | 1020.0K |
| FINGER TURN HARD | 763.5K | **333.4K** | 473.5K | 544.9K | 514.9K | 1293.3K |
| HOPPER HOP | 570.5K | 717.9K | 351.0K | **153.5K** | 236.1K | 156.7K |
| HOPPER STAND | 730.6K | 345.5K | **245.2K** | 255.2K | 333.4K | N/A |
| PENDULUM SWINGUP | 428.0K | 414.7K | 98.5K | **76.6K** | 256.0K | 193.3K |
| QUADRUPED RUN | N/A | 777.1K | 585.5K | **345.0K** | 465.0K | N/A |
| QUADRUPED WALK | 879.3K | 619.3K | 684.9K | 335.0K | **330.5K** | 573.3K |
| REACHER EASY | 711.7K | 345.5K | 354.2K | 253.8K | **205.0K** | 490.0K |
| REACHER HARD | N/A | **589.7K** | 707.0K | 827.4K | 804.9K | 1196.7K |
| WALKER RUN | 377.5K | 335.0K | 335.0K | 244.3K | **205.0K** | N/A |
| WALKER STAND | 546.7K | 121.3K | 147.0K | **120.0K** | 171.7K | 510.0K |
| WALKER WALK | 446.9K | 167.6K | 184.9K | **116.8K** | 164.5K | 490.0K |

*Table 6.* Comprehensive list of mathematical symbols and notations.

| Symbol | Description | First Appearance |
|---|---|---|
| ***State Spaces and World Model*** | | |
| $\mathcal{X}, o$ | High-dimensional observation space and raw observation $o \in \mathcal{X}$ | Sec. 3.1 |
| $\mathcal{M}, s$ | Low-dimensional latent manifold and latent state $s \in \mathcal{M}$ | Sec. 3.1 |
| $e_\psi, p_\psi$ | Observation encoder and transition world model parameterized by $\psi$ | Sec. 3.1 |
| $a_t, r_t$ | Action and extrinsic reward at time step $t$ | Sec. 3.1, Eq. 3 |
| $V_\phi, V^*$ | Parametric scalar value function field and the optimal value field | Sec. 3.1 |
| $U_{\phi_u}$ | Parametric Bellman uncertainty function | Eq. 4 |
| ***Active Causal Intervention (ACI)*** | | |
| $s'$ | Synthesized interventional anchor (destination of a latent jump) | Def. 3.1 |
| $p_{gen}$ | Learned generator distribution for initializing imagination at $s'$ | Def. 3.1 |
| $\mathcal{G}_\theta$ | Adversarial state generator parameterized by $\theta$ | Sec. 3.2 |
| $\epsilon$ | Standard Gaussian noise vector for the generator $\epsilon \sim \mathcal{N}(0, I)$ | Eq. 5, Alg. 1 |
| $\xi$ | An imagined trajectory or rollout path | Eq. 2 |
| ***Expected Free Energy and Objectives*** | | |
| $G(\pi, t)$ | Local Expected Free Energy (EFE) for policy $\pi$ at step $t$ | Def. 3.2, Eq. 1 |
| $\Psi(s, s')$ | Relay Expected Free Energy (R-EFE) functional bridging $s$ to $s'$ | Eq. 2 |
| $\mathcal{P}, \mathcal{E}$ | Pragmatic Value (goal-alignment) and Epistemic Value (info-gain) | Eq. 1 |
| $\mathcal{I}$ | Mutual Information quantifying epistemic uncertainty reduction | Eq. 1, Eq. 4 |
| $V_{RVF}$ | Relay Value Function (Pragmatic Proxy) | Eq. 3 |
| $V_{RUF}$ | Relay Uncertainty Function (Epistemic Proxy) | Eq. 4 |
| $\mathcal{L}_{mf}$ | Manifold structural consistency constraint (Cycle/Dynamics coherence) | Eq. 5 |
| $\beta, \eta$ | Weighting coefficients for Epistemic (RUF) and Pragmatic (RVF) potentials | Eq. 1, Eq. 5 |
| $\gamma$ | Pragmatic temporal discount factor | Eq. 3 |
| $\gamma^2$ | Quadratic epistemic discount factor establishing the Epistemic Horizon | Eq. 4 |
| ***Theoretical Analysis and Operators*** | | |
| $\tau_{s'}$ | First hitting time to the synthesized anchor $s'$ | Def. 4.1 |
| $\mathcal{T}_V, \mathcal{T}_U$ | Pragmatic and Epistemic Relay Operators | Def. 4.1 |
| $q_{\mathcal{G}}, q^*$ | Sampling distribution induced by $\mathcal{G}$ and optimal proposal distribution | Thm. 4.4 |
| $\mathcal{F}(\theta)$ | Fisher Information Matrix of the world model parameters $\theta$ | Prop. 4.5 |
| $\Delta V(s, s')$ | Variational Bellman Residual (Relay Advantage); $\Delta V(s) = \sup_{s'} \Delta V(s, s')$ | Thm. 4.6 |
| $\epsilon_V(s)$ | Value residual $V^*(s) - V_\phi(s)$ (Bellman error) | Thm. 4.6 |
| $\Phi$ | Conductance of the latent manifold $\mathcal{M}$ | Thm. 4.8 |
| $\nu$ | Convergence speedup ratio relative to trajectory-bound sampling | Thm. 4.8 |
| $\delta(s')$ | Hallucination error / Euclidean deviation from the true manifold $\mathcal{M}$ | Thm. 4.9 |
| $L$ | Lipschitz constant of the optimal value field $V^*$ | Thm. 4.9 |

*Table 7.* Hyperparameters for Mind Dreamer and DreamerV3 Baseline. We retain the default parameters of DreamerV3 to ensure a fair comparison, while introducing MD-specific parameters for the generator and relay potentials.

| CATEGORY | HYPERPARAMETER | VALUE |
|---|---|---|
| \multicolumn{3}{c}{*DreamerV3 Base (DMC Vision)*} | | |
| ENVIRONMENT | ACTION REPEAT | 2 |
| | IMAGE SIZE | $64 \times 64$ |
| WORLD MODEL | RSSM DETERMINISTIC / STOCHASTIC | 512 / 32 |
| | DISCRETE LATENT CATEGORIES | 32 |
| | ENCODER/DECODER CNN DEPTH | 32 |
| | MLP LAYERS / UNITS | 5 / 1024 |
| | KL FREE NATS / DYN SCALE / REP SCALE | 1.0 / 0.5 / 0.1 |
| BEHAVIOR | ACTOR / CRITIC LAYERS | 2 |
| | IMAGINATION HORIZON ($H$) | 15 |
| | DISCOUNT FACTOR ($\gamma$) / LAMBDA ($\lambda$) | 0.997 / 0.95 |
| | ACTOR ENTROPY COEFFICIENT | $3 \times 10^{-4}$ |
| OPTIMIZATION | BATCH SIZE $\times$ LENGTH | $16 \times 64$ |
| | MODEL / ACTOR / CRITIC LEARNING RATE | $1 \times 10^{-4}$ / $3 \times 10^{-5}$ / $3 \times 10^{-5}$ |
| | GRADIENT CLIPPING | 1000.0 |
| \multicolumn{3}{c}{*Mind Dreamer Specific (Ours)*} | | |
| GENERATOR $\mathcal{G}$ | MLP LAYERS (ENC / DYN) | 2 / 2 |
| | INFONCE TEMPERATURE | 0.1 |
| | INFONCE SCALE | 1.0 |
| | CYCLE CONSISTENCY SCALE ($\lambda_{mf}$) | 0.1 |
| | DYNAMICS CONSISTENCY SCALE | 0.01 |
| INFONCE WEIGHTS | REAL BUFFER NEGATIVES ($w_{real}$) | 1.0 |
| | GENERATED NEGATIVES ($w_g$) | 0.4 |
| | CROSS-BATCH NEGATIVES ($w_{cross}$) | 0.2 |
| | ELITE POOL NEGATIVES ($w_{elite}$) | 1.2 |
| RELAY POTENTIALS (RVF / RUF) | $k$-HORIZON | 15 |
| | EMA TARGET UPDATE FREQ. / FRACTION | 100 / 0.05 |
| | PRAGMATIC (SVF) WEIGHT ($w_v$) | 1.0 |
| | EPISTEMIC (SUF) WEIGHT ($w_u$) | 1.0 |

