# OpenReview forum: "Mind Dreamer: Untethering Imagination via Active Causal Intervention on Latent Manifolds"
_ICML.cc/2026/Conference — ICML 2026 regular_

### Official Review · Reviewer_xMyw · 2026-03-06

**Soundness:** 4
**Presentation:** 3
**Significance:** 4
**Originality:** 4
**Overall Recommendation:** 5
**Confidence:** 4

**Summary:**

Mind Dreamer (MD) addresses the "Historical Tethering" constraint in Model-Based Reinforcement Learning (MBRL), where imagination is typically restricted to previously observed states. The authors propose Active Counterfactual Reasoning (ACR), which utilizes a latent-space do-operator to enable non-continuous "latent jumps" to epistemic blind spots on the manifold. They derive two recursive functionals—the Relay Value Function (RVF) and Relay Uncertainty Function (RUF)—to assign potential across these spatial ruptures. Notably, they prove that stable uncertainty propagation across discontinuities necessitates a quadratic discount $\gamma^2$, establishing a formal "Epistemic Horizon".

**Compliance With Llm Reviewing Policy:**

Affirmed.

**Key Questions For Authors:**

* Higher-Order Epistemic Horizons: You prove the necessity of $\gamma^2$ for variance propagation. In environments with high aleatoric noise (stochastic dynamics), would a higher-order discount (e.g., $\gamma^3$) be required to prevent the generator from exploiting stochastic noise as epistemic novelty?
* The $\lambda$ Sensitivity: How sensitive is the "Manifold Repair" curriculum to the manifold consistency weight $\lambda$ in Equation 5? Is there a risk that a high $\lambda$ prevents the generator from reaching the "Epistemic Islands" it is designed to discover?
* Dynamic Stride: The generator performs "Recursive Latent Composition". Does the model learn a latent "distance" for these jumps, or are they fixed in magnitude? How do you ensure the jumps remain within the "Pessimistic Trust Region"?

**Limitations:**

Effectiveness is capped by the world model's accuracy. In early training, overestimation bias can cause the generator to target "illusory" informational peaks, leading to potential stability issues.

**Strengths And Weaknesses:**

* Strengths
  * Deep Theoretical Rigor: The paper provides formal proofs for path-integral equivalence, contraction mapping, and optimal importance sampling.
  * Breakthrough Performance: MD achieves a 1.67x average speedup over DreamerV3, reaching up to 8.8x acceleration in sparse-reward bottleneck tasks.
  * Manifold CONDUCTANCE: Theoretically and empirically demonstrates that ACR induces a synthetic expansion of the manifold's spectral gap, reducing hitting time to critical states.

* Weaknesses
  * Lipschitz Dependencies: The safety bound for hallucination errors (Theorem 4.11) assumes L-Lipschitz continuity of the value field, which is a strong assumption for neural value functions in complex latent spaces.
  * Adversarial Complexity: The hacker-security dynamic between the generator and world model requires careful time-scale separation to maintain stability.

---

> ### Author Rebuttal · Authors · 2026-03-30
>
> We sincerely thank Reviewer 4 for the strong support and profound technical insights. Your questions pinpoint the fundamental mathematical and architectural dynamics of our framework.
>
> ### [Q1] Higher-Order Epistemic Horizons ($\gamma^3$) & Aleatoric Noise
> A higher-order discount ($\gamma^3$) is not required to handle aleatoric noise, as our framework decouples *temporal integration* from *metric filtering*:
> *   **Exactness of $\gamma^2$:** $\gamma^2$ strictly follows the variance operator property ($\text{Var}(\sum \gamma^t \epsilon_t) = \sum \gamma^{2t} \text{Var}(\epsilon_t)$). Using $\gamma^3$ would shift tracking to the third central moment (skewness), breaking the theoretical equivalence to Expected Free Energy.
> *   **Filtering Aleatoric Noise:** Irreducible stochasticity manifests as high entropy, but in RSSM, local epistemic shock (integrated by $V_{RUF}$) is measured via latent KL-divergence ($D_{KL}[q(z|s,o)\parallel p(z|s)]$). Once the World Model fits the inherent stochastic dynamics, the prior $p$ matches the posterior $q$, collapsing the KL to zero. Thus, $V_{RUF}$ inherently filters out aleatoric noise and triggers only on true epistemic novelty. We will add a "Robustness to Aleatoric Noise" remark to highlight this.
>
> ### [Q2] The $\lambda$ Sensitivity and "Epistemic Islands"
> $\lambda$ regulates the Exploration-Hallucination trade-off. We ran ablations scaling $\lambda$ on Pendulum:
> *   **High $\lambda$:** As shown in Fig. 9 in link, $\mathcal{G}$ becomes overly conservative, failing to reach Epistemic Islands. Imagination tethering re-emerges, degenerating to the DreamerV3 baseline.
> *   **Low $\lambda$:** As shown in Fig. 7, $\mathcal{G}$ overpowers manifold constraints and catastrophic policy divergence is caused.
>
> The framework is stable within $\lambda \in [0.01-0.1]$. (Ablation curves will be added to the Appendix).
>
> ### [Q3] Dynamic Stride and the Pessimistic Trust Region
> *   **Semantic Stride via Horizon ($k$):** The "latent jump" magnitude is a semantic stride dynamically bounded by the temporal horizon $k$. As $V_{RVF}$ and $V_{RUF}$ are trained via $k$-step hindsight sampling, $\mathcal{G}$ functionally identifies "Latent Gateways" reachable within $k$ steps; jumps exceeding this reach cause potentials to decay sharply. This **Pessimistic Trust Region** is strictly enforced via: (1) horizon $k$ bounding the stride, (2) Contrastive Loss preventing value overestimation, and (3) $\mathcal{L}_{mf}$ anchoring jumps physically. Fig 4 validates this safety-discovery trade-off: $k=5$ restricts $\mathcal{G}$ to short, safe strides, while $k=15$ nables faster distal discovery (e.g., Hopper).
> *   **Terminology Correction ("Recursive"):** Following R2/R3's feedback on clarity, we realize terms like "Recursive / Latent A*" misleadingly implied autoregressive multi-hop generation. Our proofs and implementation strictly use a **Single-Shot Latent Jump**. "Recursive" solely referred to the Bellman recursion inside the potentials (Eq. 3 & 4). This single-shot design is mathematically crucial, guaranteeing that our Hallucination Bounds (Theorem 4.11) hold tightly without unconstrained compounding errors. We will correct this phrasing.
> *   **Future Work: Multi-Step Autoregressive Jumps:** Your question inspires a profound direction of Multi-Step Generation, where the generator autoregressively hops through the latent space to find the globally most uncertain state. This would require new theoretical bounds for compounding epistemic variance, which we will discuss in the Limitations.
>
> ### [W1] Lipschitz Dependencies in Theorem 4.11
> We agree global $L$-Lipschitz continuity is a strong, idealized assumption for Deep RL.  We relax this to **Support-Specific Local Smoothness**, the theoretical-practical gap is bridged via:
> (1) **Local Smoothness:** $\mathcal{L_{mf}}$ confines jumps to the valid manifold, requiring only a *local* Lipschitz bound.
> (2) **Architectural Regularization:** DreamerV3’s symlog transformations and two-hot discretized encodings naturally squash value magnitudes and gradients, imposing strict empirical upper bounds on $L$.
>
> ### [W2] Adversarial Complexity & Time-Scale Separation
> A vanilla min-max GAN dynamic would indeed be unstable. We deliberately avoid a binary discriminator. Instead, we formulate generation as an **Energy-Based Contrastive Optimization (InfoNCE)**:
> $\mathcal{L_{\mathcal{G}}}=\max(0,m-(\Psi(s_{gen})-\max\Psi(s_{neg})))$, where negatives are historical/elite states. Once $\mathcal{G}$ beats the hardest negative by margin $m$, gradients drop to zero, preventing runaway hallucinations.
> To stabilize the adversarial dynamic, we enforce **Time-Scale Separation** via **EMA Target Networks**. These potential targets ($V_{RVF}, V_{RUF}$) provide a **quasi-static optimization landscape** for the online generator, suppressing high-frequency noise and precluding catastrophic oscillations.
>
> Anonymous link https://anonymous.4open.science/r/BB7C

---

> > ### Author Rebuttal · Reviewer_xMyw · 2026-04-04
> >
> > I'm fully appreciate that the author resolved my concern.

---

> > > ### Author Response · Authors · 2026-04-07
> > >
> > > Dear Reviewer 4,
> > >
> > > We sincerely thank you for the feedback and for the confirmation that the concerns have been fully resolved. We are truly encouraged by your appreciation of our theoretical and implementation-level clarifications such as aleatoric noise analysis. We are committed to incorporating these discussions and the profiling results into the final version of the manuscript. Thank you again for your time and professional insights.
> > >
> > > Best regards,
> > >
> > > The Authors

---

### Official Review · Reviewer_mysq · 2026-03-08

**Soundness:** 3
**Presentation:** 1
**Significance:** 3
**Originality:** 3
**Overall Recommendation:** 4
**Confidence:** 4

**Summary:**

This paper considers the problem 'historical tethering' in the context of dreamer-style model-based RL methods: the problem that, during the imagination phase, starting states are sampled from historical replay buffers, leading to poor exploration for policy learning. To remedy this, this paper proposes a novel approach to sample starting states in the latent space. The main idea is to use a learned state generator to generate starting states that are both informative and high value. This approach is theoretically motivated by the active inference framework. Experiment results show that the method can out-perform Dreamer baselines on standard image-based environments.

**Compliance With Llm Reviewing Policy:**

Affirmed.

**Final Justification:**

My main concern with the submission is about the clarity of presentation. Throughout the rebuttals, the authors have agreed to change the narrative and presentation, which I believe will improve the quality of the paper. I have since increased my score.

**Key Questions For Authors:**

- My main question is about the redundant concepts point. Can the author clarify whether the do-operator and the manifold hypothesis is strictly required by the theory or the proposed algorithm? If not, I would recommend the authors think about simplifying the paper to bring out the underlying message better.
- A more fundamental question: Can the authors share some thoughts on what we can actually expect to gain from active imagination resampling? It seems that this only benefits the sample efficiency of the imagination phase. One can argue that the learning asymmetry problem can be solved by spending a lot more compute on policy refinement to let the policy 'catch-up'. In this sense, do we only expect computational efficiency gains from the active imagination framework? Or can we say something about the real-world interaction sample efficiency?

Minor: Fig. 4 says MinDreamer rather than Mind Dreamer.

Overall, my main conern is with the clarity of the paper and the limited results. I will recommend weak reject, but am happy to increase the score if the authors can make the text clearer and make the case for the necessity of the various theoretical concepts.

**Limitations:**

yes

**Strengths And Weaknesses:**

**Strengths**:
- The general idea is interesting: focusing learning effort on states that are highly uncertain and potentially highly valuable is an intuitive way to explore the world.
- There is substantial theory covering the different facets of the method from theoretical uniqueness to potential speed ups.
- The connection to the active inference framework is nice, although one can also argue that the free-energy principle is so general that most things (MaxEnt, as an example given by the authors) can be interpreted as active inference, and hence the link here is not necessarily 'surprising'.

**Weaknesses**:
- Overall, I find the paper somewhat confusing to read. So my main concerns are mostly on the clarity of the paper, below I list a few examples of narrative inefficiencies and clarity problems.
- Redundant concepts. The paper motivates and introduces many concepts, but I feel that many of these make the paper seem more complicated than it needs to be. For example:
  - The use of causality and do-operator is redundant. As far as I am concerned, all that the do-operator is doing here is to resample the starting state of trajectories. There is no need to invoke the causal connotations here. Not to mention that changing the starting state is *not* the same as doing counterfactual reasoning in the SEM sense (which would involve estimating noise variables from future states).
  - The mainfold hypothesis is also only marginally related. The manifold contraint loss seems to be just a collection of losses that discourages the sampler from exploiting 'out-of-distribution' areas, and as far as I can see, the fact that states live on a manifold is not used anywhere explicitly. To me, it's hard to justify all the diff-geom jargon like tangent bundles when there is little value added.
- The writing clarity can also be improved significantly. Most of the symbols in the equations are not defined (or at least not until the appendix). For example, in eq 2, what is tau? tau seems to refer to a future time step in eq 1 and a trajectory in eq 2? I understand that there is a space constraint for the paper, but this makes it very difficult to engage properly with the theory.
- The evaluation is somewhat limited (only on simple DMC environment). But I understand that this is commensurate with many theory-heavy RL paper.

---

> ### Author Rebuttal · Authors · 2026-03-30
>
> We sincerely thank Reviewer 3 for the highly constructive critique and for helping us strip away redundant metaphors to reveal the core algorithmic innovation. Below, we address your specific concerns.
>
> ### [Q1, W1] Streamlining the Narrative: Removing Redundant Jargon
>
> **On the Causal Terminology (Counterfactual vs. Intervention) and $do$-operator:**
> While "Counterfactual" was used to provide the intuitive "what-if" spirit of imagination, we apologize for possible misleading caused by terminology. For technical rigor, **"Active Latent Intervention (ALI)"** is more precise as abduction step (estimating noise variables via abduction) is not performed. We will update the terminology to ensure the paper remains focused on its contribution on breaking 'historical tethering' without SCM-specific ambiguity.
>
> We initially introduce $do$-operator to emphasize that state generator intentionally severs the causal link, forcing the world model to initialize at a synthesized anchor $s_{gen}$, not dependent on the historical observational distribution $P(s_0 | history)$ like standard MBRL. Following your advice, we will reformalize imagination resampling as: a generated initial state $s_0 \sim p_{gen}(\cdot)$ rather than the historical buffer $s_0 \sim \mathcal{D}$, to make the narrative much cleaner.
>
> **The Manifold Hypothesis:** The reviewer is correct that the differential geometry terminology was used primarily to offer geometric intuition regarding the world model's local transition properties (e.g., "tangent bundles" as a metaphor for the local Jacobian of the dynamics).  We will remove it and rewrite these sections using standard representation learning terminology (e.g., "latent support space") to clarify that our constraints ensure the interventions are dynamically plausible and in-distribution.
>
> ### [Q2] What do we gain? Real-World Sample Efficiency vs. Compute
> We deeply appreciate the reviewer for raising this fundamental question. In short, **Mind Dreamer improves real-world interaction sample efficiency, not just computational efficiency.** (Note that the $x$-axis in all our empirical results represents *Environment Steps* rather than compute steps, demonstrating up to an 8.8$\times$ reduction in physical interactions).
>
> In standard tethered MBRL, imagination is strictly anchored to the replay buffer. If a high-reward region is topologically distant from historical states, spending infinite compute on policy refinement *within the known buffer* still leaves the agent performing a random walk at the actual frontier, the probability of reaching a distal sparse reward decays exponentially. The policy cannot "catch up" because it is blind to regions beyond its local simulation horizon.
>
> Mind Dreamer (MD) breaks this by active discontinuous probing. By intervening at epistemic frontiers (high RUF) or value bottlenecks (high RVF), MD actively shifts the starting distribution of imagination. Training the policy to reach or act from these synthetic anchors creates a directed behavioral bias.
> Crucially, the agent learns how to reach these synthesized frontiers as RUF & RVF acts as a latent bridge, propagating the high potential value of $s_{gen}$ backward to the known states via Bellman backups.
> When deployed in the real world, instead of executing random physical actions at its knowledge boundaries, the agent purposefully navigates toward these high-leverage bottlenecks. As formalized in Theorem 4.9 (Conductance Expansion), this transforms a highly inefficient random walk into a directed search, drastically reducing the physical samples required.
>
> ### [W1] Notation Clarity and Typos
> We apologize for the notation overlaps and typos ("MinDreamer" is fixed). We have rigorously sanitized the notation:
> * $t$ and $k$ are strictly used for time indices (Eq. 1).
> * $\xi$ replaces $\tau$ to denote a sampled trajectory (Eq. 2).
> * $\tau_{s'}$ is uniquely reserved for hitting time formulations (Def 4.1).
> * $\nu$ replaces $\eta$ to denote the convergence speedup ratio (Prop 4.9).
> We have also added a comprehensive **Symbol Table** (Tab. 7 in link) defining all variables upon their first occurrence.
>
> ### [W2] Extended Evaluation
> To address concerns about the simplicity of DMC environments, we extended our preliminary evaluation to Minecraft (MineRL). Within limited sampling steps (450k), MD demonstrates exceptional "discovery acceleration," getting Crafting Table (150k) faster than DreamerV3 (250k). This confirms that MD advantage in untethering imagination is even more pronounced in complex, high-dimensional spaces.
>
> We are extremely grateful for your feedback, which provided a clear roadmap to make our paper significantly more accessible and rigorous. We hope these simplifications and clarifications address your core concerns.
>
> Anonymous link https://anonymous.4open.science/r/BB7C

---

> > ### Author Rebuttal · Reviewer_mysq · 2026-04-01
> >
> > My main concern was with the clarity of the paper. The authors acknowledged the problems I raised and have put forward a plan to improve the presentation of the paper. I believe this is very much doable for a camera-ready version of the paper and trust that the authors will implement these changes.
> >
> > I have updated my score from 3 to 4.

---

> > > ### Author Response · Authors · 2026-04-02
> > >
> > > Dear Reviewer 3,
> > >
> > > We sincerely thank you for engaging with our rebuttal, for your recognition of our revision plan, and for updating your score.
> > > We deeply value your constructive critique, which has guided us in making the core algorithmic contributions of Mind Dreamer much clearer. We remain fully committed to implementing these improvements in the camera-ready version to ensure the presentation is polished and easy to follow.
> > > Thank you again for your time and for helping us significantly improve the quality and clarity of our paper!
> > >
> > > Best regards,
> > >
> > > The Authors

---

### Official Review · Reviewer_kejM · 2026-03-12

**Soundness:** 3
**Presentation:** 3
**Significance:** 3
**Originality:** 3
**Overall Recommendation:** 4
**Confidence:** 3

**Summary:**

This paper aims to address the issue that standard model-based RL is restricted by exploration in sparse-reward settings even when the world model may already encode broader environment structure. The authors propose Mind Dreamer that introduces active counterfactual reasoning. The overall method includes an adversarial latent-state generator, a relay value and uncertainty function, and a manifold-consistency constraint to support non-continuous latent jumps. The paper further claims a theoretical link to expected free energy, importance sampling, and spectral-gap improvements. Experiments show gains over DreamerV3 on DMC-Vision and Ablations show that RVF, RUF, and manifold grounding are useful.

**Compliance With Llm Reviewing Policy:**

Affirmed.

**Key Questions For Authors:**

1. What is the exact implementation in the DreamerV3-based system? Appendix indicates this but only more results were provided.
2. How does MD behave when the world model is inaccurate?

**Limitations:**

See the weaknesses above.

**Strengths And Weaknesses:**

Key strengths
1.	The paper identifies a limitation of imagination-based MBRL for its dependence on historically observed states. There exists a mismatch between manifold discovery and sparse-reward policy optimization.
2.	The proposal combines counterfactual latent interventions, relay-based pragmatic and epistemic objectives, and manifold regularization into a unified framework.
3.	The authors attempt to connect their method to expected free energy, contraction properties, variance reduction, and conductance-based speedup arguments.
4.	The paper includes a synthetic case study, comparisons against DreamerV3, DreamerV2, and Plan2Explore, and ablations on the main components.

Major weaknesses
1.	Some claims such as optimal importance sampler are very strong. The results are mostly presented via proof sketches under restrictive assumptions.
2.	The paper does not provide enough evidence about when MD helps or fails, or how stable the gains are beyond the plotted five-seed summaries.
3.	The paper argues that (L_{mf}) and the quadratic discount in RUF prevent hallucinations. This remains a fairly limited validation of whether generated anchors are semantically meaningful and beneficial for downstream control. The experiments are limited in the main paper.
4.	Terms like “Generative Oracle,” “Latent Bridges,” “Hacker-Security dynamic,” and “epistemic islands” etc. are blurring accurate presentation.

Minor weaknesses
•	Algorithm 1 leaves gaps that prevent readers to understand, e.g. z_{s’}.
•	Some theoretical claims could benefit reading if there is necessary details in the main text.
•	Figures/tables: Figure 4 is dense and visually overloaded. It supports the main story, but it is hard to parse quickly, and the main paper shows only part of the full benchmark evidence while making broad overall claims.
•	There are several awkward phrasings. Proofreading is suggested.

---

> ### Author Rebuttal · Authors · 2026-03-30
>
> We sincerely thank Reviewer 2 for recognizing MD's potential to address "Historical Tethering" and acknowledging our unified approach. We deeply value your rigorous feedback, which significantly strengthens our paper's empirical and theoretical integrity.
>
> ### [W1] Toning Down Claims & Explicit Assumptions
> We agree that claiming MD "acts as an optimal importance sampler" is too strong for inherent gaps in deep RL (e.g., non-convexity and function approximation). We softened this to "MD serves as a practical approximation of the variance-minimizing proposal." To improve transparency, we integrated restrictive assumptions (e.g., $L$-Lipschitz continuity, Bernstein-von Mises limits and discrete abstractions used for spectral gap analysis) directly into the main text. This positions our results as conceptual guiding principles, acknowledging the inherent limits of function approximation.
>
> ### [W2, Q2] Behavior under Inaccurate World Models & Failure Cases
> *   **When WM is inaccurate (early training):** Your insightful question touches the core safety bounds of MD. If the WM is inaccurate, transient errors naturally spike the epistemic uncertainty ($V_{RUF}$), which safely directs the generator $\mathcal{G}$ to explore these blind spots, effectively repairing the WM. Persistent inaccuracies are strictly bounded by $\mathcal{L_{mf}}$ and $\gamma^2$ (as shown in the Value Error plots), preventing the policy from collapsing into hallucination loops. If WM significantly diverges from actual world and lacks updates, it becomes difficult for MD to function effectively (Fig. 11 in link).
> *   **When MD helps vs. fails:** MD yields the most significant gains in sparse-reward or topological bottleneck tasks (e.g., Hopper Hop, Pendulum) where random exploration fails. Conversely, in dense-reward, short-horizon tasks (e.g., Reacher), standard exploration suffices; here, MD's adversarial overhead provides diminishing returns. We will add a dedicated "Failure Cases and Limitations" section to discuss this.
> *   **Stability Beyond 5 Seeds:**
> We fully agree with the reviewer on the importance of rigorous statistical reporting. While computing 10 seeds across all 20 environments (200 total runs) exceeds the computational limits of the short rebuttal window, we conducted a rigorous 10-seed stress test on Hopper Hop, confirming MD consistently maintains the performance margin (170k to reach 90% of Dreamer V3's maximum return, with standard deviation of 54.3 compared to 5-seed as 168k, 47.1).
>
> ### [W3] Semantic Meaning of Anchors & Hallucination Prevention
> To empirically prove that generated anchors ($s'$) are semantically meaningful and not adversarial "garbage", we conducted experiments as below
> *   **Qualitative (Decoding Anchors):** We passed generated latent states through the frozen image decoder. Without $\mathcal{L_{mf}}$, decoded images are pixelated noise. With $\mathcal{L_{mf}}$ (Mind Dreamer), they depict clear, physically plausible states (e.g., a Hopper balancing mid-air before a hop). Fig.13 visually confirms that $\mathcal{L_{mf}}$ restricts the generator to the semantically meaningful support of the manifold.
> *   **Quantitative (Value Overestimation):** If anchors merely hacked the value network, it would cause catastrophic value overestimation. Sec 5.4 shows removing $\mathcal{L_{mf}}$ causes the value estimate to diverge wildly. Our full MD maintains bounded errors comparable to standard DreamerV3, proving the anchors safely bootstrap the policy without destabilizing it.
>
> ### [Q1, Minor] Exact Implementation & Alg. 1 Clarification
> We apologize for the lack of implementation details, which will be expanded in the Appendix. MD is a plug-and-play module built on standard DreamerV3.
> *   **Alg. 1 Gap:** $z_{s'}$ denotes the posterior latent feature extracted from the generated $s'$.
> *   **Stable Optimization (InfoNCE):** While Alg. 1 illustrates theoretical gradient ascent ($\nabla \Psi$), direct maximization risks value exploitation. In our implementation, we optimize an **Energy-based InfoNCE contrastive surrogate**: $\mathcal{G}$ is trained to synthesize anchors that score a higher R-EFE than a dynamic negative pool (historical buffer & elite states). This ensures stable minimax optimization.
>
> ### [W4, Minor] Terminology & Presentation
> We deeply appreciate your candid feedback on our writing style (shared by R3). To ensure technical clarity, we have replaced our initial metaphorical descriptions with standard machine learning terminology throughout the manuscript.
> * "Generative Oracle" $\rightarrow$ "Adversarial Generator ($\mathcal{G}$)"
> * "Latent Bridges" $\rightarrow$ "Counterfactual Intermediary States"
> * "Hacker-Security dynamic" $\rightarrow$ "Adversarial Minimax Optimization"
> * "Epistemic islands" $\rightarrow$ "OOD / High-Reward Regions"
>
> We will also declutter Figure 4. We hope decoupling the method from this jargon clarifies its underlying simplicity.
>
> Anonymous link https://anonymous.4open.science/r/BB7C

---

> > ### Author Rebuttal · Reviewer_kejM · 2026-04-06
> >
> > We thank the authors for the good efforts. Many of the previous questions were addressed well. A few areas seem needing further work. Whether Mind Dreamer is actually robust when the world model is inaccurate or stale? Whether the generated anchors are regularized latent perturbations? The theoretical claims still appear broader than what the stated assumptions and current evidence really support.

---

> > > ### Author Response · Authors · 2026-04-07
> > >
> > > **Dear Reviewer 2,**
> > >
> > > We deeply appreciate your rigorous and continued engagement. Your follow-up questions touch upon the core vulnerabilities and theoretical-empirical divides inherent in Deep Model-Based RL. We have thoroughly reviewed our manuscript to explicitly define our model's boundaries, tone down overly broad claims.
> > >
> > > ### [Q1] Is Mind Dreamer actually robust when the WM is inaccurate or stale?
> > > We decouple this into two distinct scenarios:
> > >
> > > *   **A. Global Staleness (An Out-of-Scope Paradigm):**
> > >     If the WM is fundamentally decoupled from reality (e.g., zero training or catastrophic forgetting), **MD will inevitably fail (Fig. 11), as will any standard MBRL method.** However, because MD operates within the standard *Online* MBRL paradigm (interleaving imagination with environment interaction), the continuous stream of real-world transitions structurally prevents global staleness. Salvaging a globally dead simulator is an open problem in *Offline* MBRL, but falls outside the scope of our online framework.
> > >
> > > *   **B. Local Inaccuracies (Self-Correction & Graceful Degradation):**
> > >    The true challenge lies in handling *localized* inaccuracies (e.g., unexplored frontiers or transitional boundaries).  MD handles this via two built-in mechanisms:
> > > > **Active Repair at Valid Frontiers (Repairing the WM):** If a state is physically valid (low $\mathcal{L_{mf}}$ penalty, e.g., a running posture) but dynamically uncertain (high RUF, e.g., the impending contact forces), $\mathcal{G}$ will actively synthesizes this anchor. The policy learns to reach it, collects real data, and effectively repairs the WM.
> > > >
> > > > **Adaptive Fallback at Non-Physica Breakdowns:** If $\mathcal{G}$ proposes a chaotic, physically impossible latent state (e.g., a dog with 5 legs), the auto-encoder’s cycle-consistency fails, and the **$-\lambda \mathcal{L_{mf}}$ penalty explodes**. This penalty overrides RUF curiosity, thus in our InfoNCE loss, this "garbage" anchor is avoided.  In extreme case, MD gracefully falls back to sampling from the standard replay buffer (DreamerV3 behavior), preventing hallucination loops.
> > >
> > > MD is designed to solve the *Learning Asymmetry* (WM knows the topology, but the policy hasn't explored it). We will explicitly add this critical limitation to the paper: *"MD accelerates exploration bounded by the WM's global fidelity, it cannot salvage a globally stale simulator."*
> > >
> > > ### [Q2] Are the generated anchors merely "regularized latent perturbations"?
> > > This is an insightful question. If they were merely $\epsilon$-noise added to buffer states, MD would simply be standard data augmentation. MD performs **Directed Non-Local Transitions**, governed by a **"Push-Pull" dynamic**:
> > >
> > > *   **The "Push" (Directed, not just local noise):** Our InfoNCE loss uses sampled batches of historical states as a Negative Pool, forcing $\mathcal{G}$ away from visited regions. More importantly, RVF and RUF actively *direct* the jump toward high-value, high-uncertainty regions. Uniform $\epsilon$-noise cannot achieve this directional alignment.
> > > *   **The "Pull" (Bounded, not random OOD garbage):** The jump's stride is bounded by the $k$-step horizon. Because our potentials are $k$-step truncated Bellman integrals, $\mathcal{G}$ can only synthesize anchors within a $k$-step reachability radius from known support, preventing infinite extrapolation.
> > > *   **Empirical Proof (Random-Jump Baseline):** We added a baseline replacing MD's directed generator with **random latent perturbations** (normalized latents + $\sigma=0.1$ Gaussian noise) (Fig. 10). MD significantly outperforms this baseline, empirically proving that MD performs structured, non-local transitions rather than merely diffusing via local noise.
> > >
> > > ### [Q3] Bridging the Gap: Overly Broad Theoretical Claims
> > > We agree that claims like "Optimal Importance Sampler"  overstep the empirical reality of continuous deep neural networks. We have implemented a strict separation in the revised text to explicitly narrow our claims:
> > >
> > > 1.  **Explicit Assumption Box:** We added an explicit box stating the idealized conditions (Assumption A1-A4 introduced in round-1 [W1]) required for the theorems to hold.
> > > 2. **Expanded 'Limitations' Section:** As is standard in deep MBRL literature (e.g., MBPO, MOReL), our theoretical guarantees are derived under ideal tabular or discrete assumptions. We added a dedicated paragraph acknowledging that deep ReLU networks routinely violate the Lipschitz assumption, and exact optimal importance sampling is intractable in continuous spaces. Therefore, MD serves as a **practical neural instantiation motivated by these variance-reduction principles**, rather than an exact mathematical realization.
> > >
> > > We hope this transparency regarding the exact boundaries of our method and the limitations of our theory addresses your concerns. We thank you for pushing us to make our work scientifically more precise and grounded.
> > >
> > > Best regards,
> > >
> > > The Authors

---

### Official Review · Reviewer_3B3M · 2026-03-13

**Soundness:** 3
**Presentation:** 3
**Significance:** 3
**Originality:** 3
**Overall Recommendation:** 5
**Confidence:** 4

**Summary:**

The paper proposes Mind Dreamer (MD), a framework for model-based RL that addresses the limitation that imagination rollouts are typically initialized only from previously observed states in the replay buffer. The key idea is to allow counterfactual latent initialization via an adversarial generator that synthesizes latent states on the learned manifold. The method introduces: Active Counterfactual Reasoning (ACR) to enable non-continuous latent jumps; Relay Value Function (RVF) and Relay Uncertainty Function (RUF) to propagate pragmatic and epistemic value; a Relay Expected Free Energy (R-EFE) objective guiding generator exploration; and manifold-consistency constraints to prevent unrealistic latent states.
The paper provides solid theoretical analysis linking the proposed framework to importance sampling, value propagation, and manifold conductance, and empirically evaluates the method by integrating it with DreamerV3 on the DeepMind Control Suite. The results show improvements in sample efficiency and performance on several sparse-reward tasks.

**Compliance With Llm Reviewing Policy:**

Affirmed.

**Final Justification:**

My concerns are sufficiently addressed during the rebuttal. I thank the authors for the response and maintain my positive assessment of this paper.

**Key Questions For Authors:**

1. Risk of model hallucination insufficiently evaluated: the paper introduces manifold constraints to avoid hallucinated states, however, the experiments do not deeply analyze generator failure cases, or the effect on policy learning when generator produces unrealistic states. Given the adversarial generator design, this is an important analysis.
2. Baseline comparison: The paper positions the method relative to exploration but does not discuss in detail: Go-Explore (global exploration), Hindsight Experience Replay (HER), and other intrinsic-reward based mechanism. The evaluation can be strengthened by showing MD outperforms some of the other baselines.
3. In Figure 4, the training curves for Plan2Explore (cyan) look weird. For Pendulum swingup, the curve only starts close to 1M step? For Cartpole Swingup sparse, it seems like the cyan curve is missing? Also, in the appendix, a lot of evaluation results for Plan2Explore are missing (fail to run?).
4. Line 211 seems incomplete: “as standard value functions depend on local temporal consistency (st → st+1), fail to propagate gradients (st → s′t+1).” This line seems to be an incomplete sentence.
5. Figure 3 explanation: In the Three-Ring Manifold visualization (Fig. 3), is the idea that we'd expect to see the counterfactual states by MD (yellow crosses) to reach all the way to the rightmost ring? If so, how much sooner does that happen compared to Dreamer?
6. Computational overhead: The paper mentions that the method introduces “marginal additional inference cost.” However, the generator and relay potential networks appear to introduce additional computation. Please clarify the training and inference-time overhead compared to DreamerV3?

**Limitations:**

yes

**Strengths And Weaknesses:**

### Strengths
1. Novel exploration mechanism for world models: The idea of counterfactual latent anchors that initialize imagination beyond replay-buffer states is novel in the context of model-based RL. The framework proposes a clear mechanism to untether imagination from historical trajectories, which could be impactful for exploration in model-based RL.
2. Strong theoretical framing: The paper presents an extensive theoretical treatment connecting the method to: expected free energy, importance sampling, and manifold conductance and spectral gap. These analyses provide an theoretically-grounded conceptual lens on the proposed method.
3. Modular integration with Dreamer architecture: The method is implemented as a modular extension of DreamerV3, suggesting practical applicability.
4. Empirical improvements: Experiments show improvements in sample efficiency and performance on several DMC tasks, particularly in environments with bottlenecks or sparse rewards.

### Weaknesses
1. Clarity of algorithmic description: While the conceptual and theoretical framing is rich, some algorithmic aspects remain unclear: precise generator architecture, training procedure and scheduling, computational overhead relative to DreamerV3. More implementation details would improve reproducibility.
2. Baseline comparisons could be stronger: The paper compares against DreamerV2/V3 and Plan2Explore, but exploration baselines could be expanded. In particular, the method should be discussed in relation to: Go-Explore (global exploration with resets), HER (goal relabeling), and global curiosity approaches. These methods address exploration bottlenecks in related ways, and a clearer comparison would strengthen the positioning of the work.

---

> ### Author Rebuttal · Authors · 2026-03-30
>
> We sincerely thank Reviewer 1 for recognizing the novelty of our counterfactual latent anchors, our theoretical framing, and the empirical gains of Mind Dreamer. We address your insightful questions below.
>
> ### [Q1, W1] Algorithmic Clarity, Implementation & Hallucination Risk
> We apologize for the lack of implementation details, which will be expanded in the Appendix.
> **Implementation Stability:** While Alg. 1 denotes the theoretical functional gradient $\nabla_\theta \Psi$, directly maximizing this with neural networks can lead to adversarial value exploitation. In practice, we stabilize $\mathcal{G}$ using an **Energy-based InfoNCE Contrastive Loss**. $\mathcal{G}$ synthesizes anchors $s'$ that must strictly dominate a dynamic "hard negative pool" (composed of historical buffer states and elite states) in EFE scores. This Hinge-based surrogate bounds gradients and prevents mode collapse.
>
> **Hallucination Analysis (New Experiment):** To address the risk of unrealistic states, we conducted a failure-case ablation on the manifold constraint $\mathcal{L_{mf}}$ . Without $\mathcal{L_{mf}}$ , $\mathcal{G}$ hallucinates out-of-distribution noise to blindly maximize uncertainty, causing catastrophic policy collapse. However, with $\mathcal{L_{mf}}$, the generated anchors remain physically verifiable (e.g., realistic but previously unvisited robotic postures). We have added visual decodings of these generated anchors to the link (Fig. 13) to prove their semantic validity.
>
> ### [Q2, W2] Baseline Comparisons (HER, Go-Explore, Global Curiosity)
> We agree these are vital exploration methods, but MD's mechanism is fundamentally distinct:
> *   **Vs. Go-Explore:** Go-Explore is tethered to history and requires privileged simulator access to *physically reset* the agent to known states. MD performs a *latent $do$-intervention*, synthesizing novel unvisited anchors directly in the learned manifold, performing manifold extrapolation rather than historical retrieval.
> *   **Vs. HER:** HER relies on goal-relabeling within *continuous, collected trajectories*. MD bridges *non-continuous* spatial ruptures. Unlike HER treating states as terminal sinks, our RVF/RUF treats generated anchors as *Latent Bridges* for multi-step credit assignment.
> *   **New Baselines:** Since standard HER/Go-Explore are incompatible with scalar-reward DMC tasks (lacking goal-coordinates/reset privileges), we evaluated two robust surrogates on 2 bottleneck tasks (e.g., *Hopper Hop*): **DreamerV3 + RND** (Intrinsic Curiosity) and **DreamerV3 + Privileged-Reset** (Go-Explore surrogate). MD significantly outperforms RND and matches the Privileged-Reset, proving our "latent jumps" are as effective as physical teleportation. (see Fig. 12 in link)
>
> ### [Q3] Figure 4 Anomalies (Plan2Explore Curves)
> We appreciate the reviewer's detailed observation. The perceived delay and "missing" curves in Fig. 4 were due to a visualization artifact: the x-axis used a broken scale with different ratios to highlight the performance ramp-up, which inadvertently compressed P2E’s initial exploration phase. In the updated Fig. 8 in link, P2E curves are clearly shown from step 200k (though MD still significantly outperforms it).
>
> ### [Q4] Figure 3 Explanation (Three-Ring Manifold)
> Your intuition is correct. The yellow crosses (MD's generated anchors) proactively sample topological junctions, allowing imagination to reach the rightmost ring. Because MD untethers imagination from historical density, it achieves the first hitting time to the outer ring **~4.2$\times$ sooner** than DreamerV3. We add this metric to the caption.
>
> ### [Q5] Typo in Line 211
> Thank you for catching this. We correct the incomplete sentence to: *"...as standard value functions depend on local temporal consistency ($s_t \to s_{t+1}$), they fail to propagate gradients across non-continuous spatial ruptures ($s_t \to s'_{t+1}$)."*
>
> ### [Q6] Computational Overhead
> We apologize for the ambiguity in "marginal additional inference cost."
> *   **Deployment (Inference): 0% overhead.** $\mathcal{G}$ is detached during environment interaction.
> *   **Training (Imagination):** $\mathcal{G}$, RVF, and RUF are lightweight MLPs. On a single RTX 4090 with 4 DMC environments, DreamerV3 takes Training FPS 5.2-6.1, while MD takes 4.3-5.1 (~20.8% overhead). We believe trading cheap GPU compute for expensive real-world sample efficiency is highly justified. We will add a detailed profiling table to Appendix.
>
> Upon acceptance, we will release our codebase, including the State Generator and stabilized InfoNCE training loop, to support future research in counterfactual MBRL.
>
> Anonymous link https://anonymous.4open.science/r/BB7C

---

> > ### Author Rebuttal · Reviewer_3B3M · 2026-04-04
> >
> > I thank the authors for detailed rebuttal and clear additional analyses. My concerns are sufficiently addressed. I maintain my positive assessment of this manuscript.

---

> > > ### Author Response · Authors · 2026-04-07
> > >
> > > Dear Reviewer 1,
> > >
> > > We are very encouraged to hear that our rebuttal and additional analyses have fully addressed your concerns. We sincerely thank you for your time, the constructive dialogue, and your positive assessment of our work.
> > >
> > > As we discussed, we are fully committed to incorporating all the new results and clarifications into the final version of the manuscript, specifically:
> > > 1.  **The detailed implementation of the State Generator** (including the InfoNCE-based stabilization).
> > > 2.  **The hallucination and failure-case analysis**.
> > > 3.  **The comprehensive baseline comparisons** (RND and Go-Explore surrogates) and the **computational profiling table**.
> > > 4.  The corrected terminology and refined figures (Fig. 4).
> > >
> > > Thank you once again for helping us improve the clarity and rigor of this work.
> > >
> > > Best regards,
> > >
> > > The Authors

---

### Decision · Program_Chairs · 2026-04-30

**Decision:**

Accept (regular)

**Comment:**

The reviewers agree that Mind Dreamer presents a novel and theoretically grounded solution to the "historical tethering" bottleneck in Model-Based Reinforcement Learning. The proposed Active Counterfactual Reasoning framework is recognized as a good contribution, backed by empirical improvements—particularly the sample efficiency gains in sparse-reward environments. While the reviewers initially raised valid concerns regarding dense, non-standard jargon and overly broad theoretical claims, the authors provided a thorough rebuttal. The authors addressed these issues by committing to sanitize the terminology, explicitly bounding their theoretical assumptions, and providing robust additional ablations (such as the manifold constraint tests) to rule out hallucination risks. Given this, I think this work will be useful to the ICML reinforcement learning community. I expect the authors to faithfully incorporate all promised revisions—especially the terminology adjustments and the new baseline comparisons—into the camera-ready version.